# Anderson impurities in edge states with nonlinear and dissipative perturbations

**Vinayak M. Kulkarni⋆ and N. S. Vidhyadhiraja**

Theoretical Sciences Unit, Jawaharlal Nehru Centre for Advanced
Scientific Research, Bangalore 560064, India

⋆ vmkphysimath@gmail.com

## Abstract

We show that exceptional points (EPs) and non-Hermitian behavior can emerge dynamically in impurity models with Hermitian microscopic origins. Using perturbative renormalization group (RG) analysis, Fock-space diagonalization, and spin-spin relaxation time calculations, we demonstrate that nonlinear (NL) dispersion and anisotropic pseudochiral ($\mathcal{PC}$) interactions generate complex fixed points and spectral defectiveness. The effective Kondo model features a square-root RG invariant linking planar and longitudinal Dzyaloshinskii-Moriya (DM) couplings, driving the onset of EPs. Our analysis reveals dissipative fixed points stabilized by an emergent Lie algebra structure and a scaling collapse in relaxation dynamics. Across both single- and two-impurity extensions, we identify a universal "sign-reversion" (SR) regime near critical NL coupling, where anisotropy preserves $\mathcal{PC}$ symmetry and SR serves as a signature of non-Hermitian flow. These results establish a new class of non-Hermitian criticality generated through RG evolution in otherwise Hermitian systems.

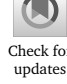

# 1 Introduction

Exceptional points (EPs) are spectral degeneracies of non-Hermitian (NH) systems where both eigenvalues and eigenvectors coalesce, rendering the Hamiltonian defective and non-diagonalizable [1–3]. These degeneracies lie at the heart of NH phenomena such as non-reciprocal dynamics, enhanced sensitivity, and topological transitions. While EPs are well understood in open systems with engineered gain and loss [4], their emergence in *closed*, interacting quantum systems remains an active frontier.

In this work, we show that EPs can arise dynamically within a fully Hermitian microscopic model through renormalization group (RG) flow. Specifically, NL bath dispersions and ADM impurity interactions conspire to generate RG invariants with a square-root structure. This structure governs the flow of couplings and induces complex-valued fixed points, signaling the emergence of NH behavior—even though the original Hamiltonian is entirely Hermitian.

Our approach is distinct from prior studies that explicitly introduce non-Hermitian terms via boundary conditions [5,6], Lindbladian dynamics [7], or symmetry constraints [8]. Instead, we begin with a Hermitian Anderson impurity model in a bath with $C_3$ symmetry and spin-dependent nonlinear dispersion. Upon projection to the low-energy sector, we obtain an effective Kondo model featuring anisotropic Dzyaloshinskii-Moriya (ADM) and potential scattering terms. We demonstrate that these generate non-Hermitian $\mathcal{PC}$-symmetric operators under RG, leading to fixed points with distinct topological signatures.

Our work is partly inspired by Bethe Ansatz studies [9] and recent work on NH sensitivity and condition numbers [10]. We build on these insights using Poorman scaling, Fock-space diagonalization, and transport calculations to show how NL and anisotropies drive the formation of EPs.

We also explore a two-impurity generalization, where we uncover a novel "Sign Reversion" (SR) regime at critical values of the NL coupling $J_{k^3}$. Near these points, ADM terms reverse direction under RG flow, and spin relaxation time exhibits a universal scaling collapse, indicating emergent dissipative criticality.

*The key message of this work is that RG flow in Hermitian systems can dynamically generate EPs and NH criticality through NL, ADM interactions. This opens a new route to exceptional physics in closed quantum impurity systems.*

## 2 Model and formalism

We begin with a bath model considered in the context of topological insulators [11], and the Hamiltonian reads as:

$$\hat{H} = \alpha k_{\parallel}^2 \mathbb{I} + \beta k_{\parallel}^3 \cos(3\theta)\sigma_z + i\lambda \hat{z} \cdot (\vec{k} \times \vec{\sigma}), \tag{1}$$

where $\vec{k} = k_x \hat{i} + k_y \hat{j} + k_{\parallel} \hat{k}$ and $\vec{\sigma} = \sigma_x \hat{i} + \sigma_y \hat{j} + \sigma_z \hat{k}$, with $\{\sigma_x, \sigma_y, \sigma_z\}$ being Pauli matrices. The parameters $\alpha$, $\beta$, and $\lambda$ correspond to quadratic, cubic, and linear couplings in the topological system, respectively. The second quantized form of the Hamiltonian for spinful fermions is $H = \sum_k \psi_k^\dagger \hat{H} \psi_k$. Where the basis vector is $\psi_k^\dagger = \begin{pmatrix} c_{k\uparrow}^\dagger & c_{k\downarrow}^\dagger \end{pmatrix}$. Introducing an impurity with on-site interaction generally regarded as the single impurity Anderson model (SIAM) and hybridization with the edge states leads to the Hamiltonian:

$$H_{\text{SIAM}} = \psi^\dagger \hat{H} \psi + H_d + \sum_{k\sigma} V_k \left( c_{k\sigma}^\dagger d_\sigma + \text{h.c.} \right), \tag{2}$$

where $H_d$ is defined as $H_d = \sum_\sigma \epsilon_d d_\sigma^\dagger d_\sigma + U n_{d\uparrow} n_{d\downarrow}$. To simplify the bath Hamiltonian, we use the parametrization $k_x \pm i k_y = k_{\parallel} e^{\pm i\theta}$ and apply a $k$-dependent unitary operation on the bath operators, preserving canonical relations. This results in a NL $k$-dependent coefficient as shown below,

$$\mathcal{U} = \frac{1}{\sqrt{\mathcal{N}}} \begin{pmatrix} e^{i\frac{\theta}{2}} \alpha_{k1} & -i e^{i\frac{\theta}{2}} \alpha_{k2} \\ e^{-i\frac{\theta}{2}} \alpha_{k2} & i e^{-i\frac{\theta}{2}} \alpha_{k1} \end{pmatrix}. \tag{3}$$

In above equation (3) coefficients $\alpha_{k1} = \sqrt{\Delta + \beta k^3 \cos 3\theta}$ and $\alpha_{k2} = \sqrt{\Delta - \beta k^3 \cos 3\theta}$. And the normalization constant is $\mathcal{N} = |\alpha_{k1}|^2 + |\alpha_{k2}|^2 = \Delta$, where

$$\Delta = \sqrt{\beta^2 k^6 \cos^2 3\theta + \lambda^2 k^2} = \beta k^3 \gamma \cos 3\theta,$$

with $\gamma = \sqrt{1 + \frac{\lambda^2}{\beta^2 k^4 \cos^2 3\theta}}$. In the limit $\lambda \to 0$ or $\beta \to \infty$, the $\gamma$ will tend to unity. In equation (3) choice for $\alpha_{k2}$ can be $\sqrt{\beta k^3 \cos 3\theta - \Delta}$ or $i\sqrt{\Delta - \beta k^3 \cos 3\theta}$. Both forms diagonalize the bath Hamiltonian having two chiral bands with the eigenenergies $\epsilon_{k\zeta} = \alpha k^2 + \zeta \Delta$ where the emergent quantum numbers $\zeta = \pm$ represent chiral bands and it's eigenvalues are shown in figure 1. This non-interacting spectrum is similar to Rashba study [12] except from NL term which serves as additional parameter to tune band touching to gapped phase in the bath. This unitary transformation rotates the bath operators as $\tilde{\psi}_k = \mathcal{U}_k \psi_k$. Such $k$-dependent operations are used in the case of Weyl multiplicity [13, 14] having different NL dispersion as $(k_x \pm i k_y)^3$.

Basis after rotation $\tilde{\psi} = \begin{pmatrix} c_{k+} \\ c_{k-} \end{pmatrix} = \mathcal{U}\psi$ can be shown as,

$$\begin{aligned} c_{k+} &= \frac{1}{\sqrt{\beta\gamma k^3 \cos 3\theta}} \left( e^{-i\frac{\theta}{2}} \alpha_{k1} c_{k\uparrow} + e^{i\frac{\theta}{2}} \alpha_{k2} c_{k\downarrow} \right), \\ c_{k-} &= \frac{1}{\sqrt{\beta\gamma k^3 \cos 3\theta}} \left( i e^{i\frac{\theta}{2}} \alpha_{k2} c_{k\uparrow} - i e^{i\frac{\theta}{2}} \alpha_{k1} c_{k\downarrow} \right). \end{aligned} \tag{4}$$

Using $\psi = \mathcal{U}^{-1} \tilde{\psi}$ to express the original spin basis in terms of these new chiral basis,

$$\begin{aligned} c_{k\uparrow} &= \frac{1}{\sqrt{\beta\gamma k^3 \cos 3\theta}} \left( e^{i\frac{\theta}{2}} \alpha_{k1} c_{k+} - i e^{i\frac{\theta}{2}} \alpha_{k2} c_{k-} \right), \\ c_{k\downarrow} &= \frac{1}{\sqrt{\beta\gamma k^3 \cos 3\theta}} \left( e^{-i\frac{\theta}{2}} \alpha_{k2} c_{k+} + i e^{-i\frac{\theta}{2}} \alpha_{k1} c_{k-} \right). \end{aligned} \tag{5}$$

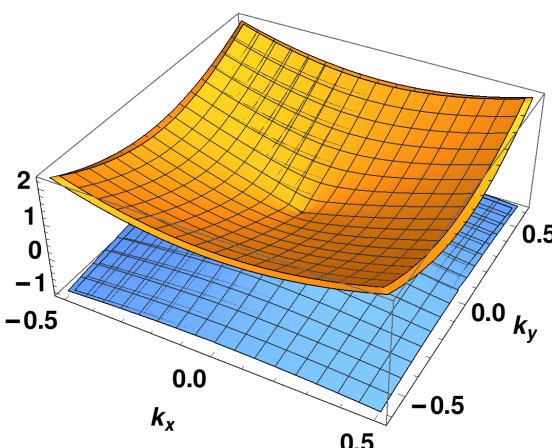

Figure 1: The eigenvalues of the diagonalized model in equation (1) around the low k points represent the emergent chiral bands. In this case, we set $\beta = 0.3$, $\lambda = 1.0$, and $\theta = \frac{\pi}{3}$. We observed that larger $\beta$ values flatten the bands.

In new basis the hybridization transforms as follows:

$$
\begin{aligned}
\tilde{H}^+_{hyb} &= \sum_{k\theta} \tilde{V}^\theta_k \left( e^{-i\frac{\theta}{2}} \alpha_{k1} c^\dagger_{k+} d_\uparrow + \text{h.c.} \right) + \sum_{k\theta} \tilde{V}^\theta_k \left( e^{i\frac{\theta}{2}} \alpha_{k2} c^\dagger_{k+} d_\downarrow + \text{h.c.} \right), \\
\tilde{H}^-_{hyb} &= \sum_{k\theta} \tilde{V}^\theta_k \left( ie^{-i\frac{\theta}{2}} \alpha_{k2} c^\dagger_{k-} d_\uparrow + \text{h.c.} \right) + \sum_{k\theta} \tilde{V}^\theta_k \left( -ie^{i\frac{\theta}{2}} \alpha_{k1} c^\dagger_{k-} d_\downarrow + \text{h.c.} \right).
\end{aligned}
\tag{6}
$$

In the above equation (6) $\tilde{V}^\theta_k = \frac{V_k}{\sqrt{\beta\gamma k^3 \cos 3\theta}}$ has the dependence $k, \theta$. A square root momentum dependent hybridization found in Rashba coupling studies [15–18]. After the unitary rotation, we obtain the renormalized SIAM as follows:

$$
\tilde{H} = \sum_{k\zeta} \epsilon_{k\zeta} c^\dagger_{k\zeta} c_{k\zeta} + H_d + \tilde{H}^+_{hyb} + \tilde{H}^-_{hyb}.
\tag{7}
$$

## Inherent symmetries

While a detailed discussion on emergent NH symmetries is deferred to later sections, we briefly note certain inherent symmetries in the Hermitian model(7). For $U = 0$, the Hamiltonian exhibits an anticommutation relation $\{\tilde{H}, \eta\} = 0$, with $\eta = \sigma_x \otimes \sigma_x$. In operator form, this reads $\psi^\dagger \eta \psi$, where

$$
\psi^\dagger = \begin{pmatrix} c^\dagger_+ & c^\dagger_- & d^\dagger_\uparrow & d^\dagger_\downarrow \end{pmatrix},
$$

as discussed in Ref. [12]. More generally, NH systems involving topology admit extended symmetry classes [19]:

$$
\begin{aligned}
\eta H \eta^{-1} &= H^\dagger, &&(\mathcal{PT} \text{ symmetry}), \\
\eta H \eta^{-1} &= -H^\dagger, &&(\mathcal{PC} \text{ symmetry}).
\end{aligned}
\tag{8}
$$

In the Hermitian case ($H = H^\dagger$), these reduce to $[H, \eta] = 0$ and $\{H, \eta\} = 0$, respectively. These symmetries inform how exceptional points can arise from complex couplings without breaking symmetry.

# 3 Effective model derivation and Poorman scaling

To project this model in equation (7) onto the impurity subspace, we use projection operator method [20]. The projection operators are defined as $P_0 = (1 - n_\uparrow)(1 - n_\downarrow)$, $P_1 = \sum_{\sigma=\uparrow,\downarrow} n_\sigma(1 - n_{\bar{\sigma}})$, and $P_2 = n_\uparrow n_\downarrow$ for unoccupied, singly occupied, and doubly occupied states, respectively. Using these projections, we derive the components of the effective model as $H_{\eta\eta'} = P_\eta H P_{\eta'}$ which is detailed in Appendix A.

$$H^\eta_{eff} = H_{\eta\eta} + \sum_{\substack{\eta' \neq \eta = 0,1,2 \\ \zeta,\zeta'=\pm}} H^\zeta_{\eta\eta'} \frac{1}{E - H_{\eta'\eta'}} H^{\zeta'}_{\eta'\eta}. \tag{9}$$

The singly occupied subspace Hamiltonian is the low-energy effective model for the Kondo regime. We show here for such a topological bath one gets an effective model which has emergent ADM interactions. We adopt a specific convention for representing couplings. We utilize vector notation to denote couplings, while pseudo-spin of bath operators are represented using bold symbols. This notation allows us to distinguish between different types of operators.

$$H_{eff} = H_0 + \sum_{kk'} J_0 s \cdot \boldsymbol{S}_{kk'} + i \sum_{kk'} \vec{J}_{k^3} \cdot (s \times \boldsymbol{S}_{kk'}) + i \sum_{kk'} \vec{J}_k \cdot (s \times \boldsymbol{S}_{kk'}). \tag{10}$$

In equation (10), $H_0 = \sum_{k\zeta} \epsilon_{k\zeta} c^\dagger_{k\zeta} c_{k\zeta}$ and the operator $\boldsymbol{S}_{kk'}$ represents the Abrikosov pseudo-spin for conduction electrons, while $s$ corresponds to the impurity spin. The couplings are denoted as $J_0 = (\alpha_{k1}\alpha_{k'1} + \alpha_{k2}\alpha_{k'2})M^{\theta,\theta'}_{kk'\zeta}$, $\vec{J}_{k^3} = (\alpha_{k1}\alpha_{k'1} - \alpha_{k2}\alpha_{k'2})M^{\theta,\theta'}_{kk'\zeta}\hat{z}$ and $\vec{J}_k = \alpha_{k1}\alpha_{k'2}M^{\theta,\theta'}_{kk'\zeta}(\hat{x}+\hat{y})$. Here, $M^{\theta\theta'}_{kk'\zeta} = \tilde{V}^\theta_k \tilde{V}^{\theta'}_{k'}\left(\frac{1}{\epsilon_{k'\zeta}-\epsilon_d} + \frac{1}{\epsilon_d+U-\epsilon_{k\zeta}}\right)$ and $\tilde{V}^{\theta'}_{k'} = \frac{V_{k'}}{\sqrt{\beta\gamma(k')^3\cos 3\theta'}}$. The matrix elements in this problem in general depend on polar angles and momenta are derived in Appendix A. The NL dispersion introduces cross-product terms in z component and linear term will introduce x,y components of DM. Poorman RG will yield following equations(EQ's),

$$\frac{dJ_0}{dl} = J_0^2 + J_{k^3}J_k + J_{k^3}^2 + J_k^2, \qquad \frac{dJ_{k^3}}{dl} = J_k^2 + J_0 J_{k^3}, \qquad \frac{dJ_k}{dl} = J_0 J_k. \tag{11}$$

The appendix B.1 contains a detailed derivation and the complete solution to the RG EQs (11). One of the solutions, obtained by eliminating $J_0$ in the EQs $\frac{dJ_k}{dl}$ and $\frac{dJ_{k^3}}{dl}$, is given by the EQ $J_k^2 - J_k = mJ_{k^3}$, where $m$ can be a positive or negative value. The roots of this solution are expressed as $J_k = \frac{1}{2} \pm \frac{1}{2}\sqrt{1+4mJ_{k^3}}$. In the low-energy effective model, it plays a significant role in causing exceptional points, which will be discussed in detail in the subsequent section, along with the diagonalization.

# 4 Emergence of non-Hermiticity and exceptional points

We performed poorman RG on Hamiltonian as in the EQ (11) and found the invariants $\tilde{J}_0$, $\tilde{J}_k$ as shown in Appendix B. Here we analyse the local Hamiltonian symmetry properties and its eigenvalue spectrum using the RG invariants for couplings and varying the $J_{k^3}$.

$$\tilde{H} = \sum_{kk'} \tilde{J}_0 s \cdot \boldsymbol{S}_{kk'} + i \sum_{kk'} \vec{\tilde{J}}_{k^3} \cdot (s \times \boldsymbol{S}_{kk'}) + i \sum_{kk'} \vec{\tilde{J}}_k \cdot (s \times \boldsymbol{S}_{kk'}). \tag{12}$$

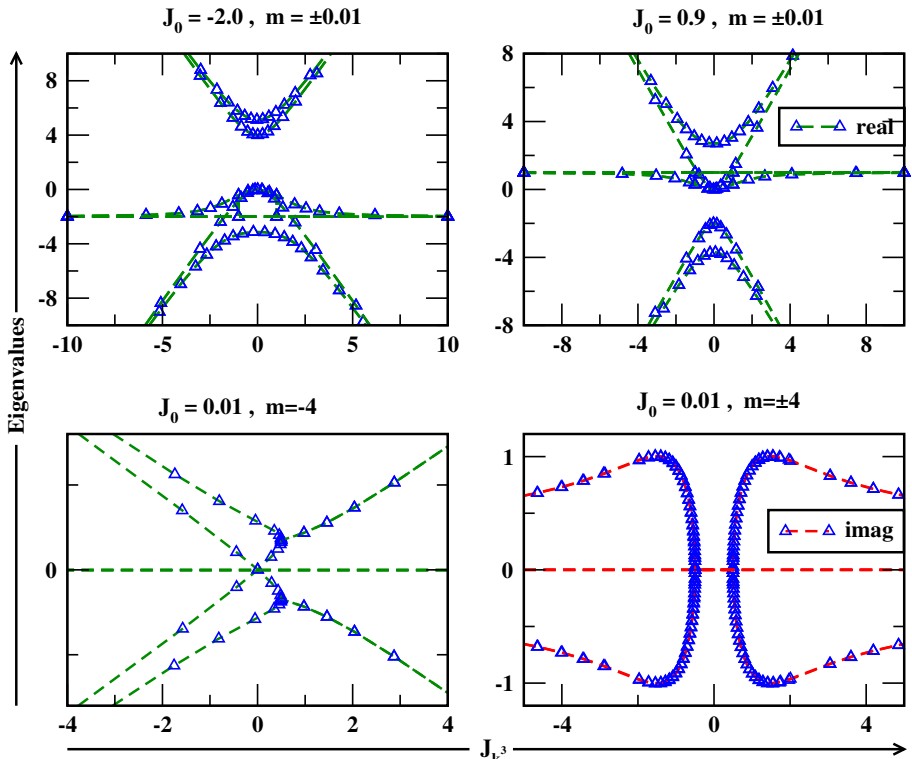

Figure 2: Eigenvalue spectra of the model (18) for small RG invariant $m = \pm 0.01$ (top row) and large $m = \pm 4$ (bottom row). Top panels show the real parts of the spectrum for $J_0 = -2.0$ (left) and $J_0 = 0.9$ (right), with green dashed lines denoting analytical results. The bottom left shows the real spectrum at $J_0 = 0.01, m = -4$, while the bottom right shows the imaginary part for $J_0 = 0.01, m = \pm 4$, with red dashed lines showing EP's features. EP's emerge near $J_{k^3} = \pm 0.5$, and strong splitting of imaginary branches reflects enhanced NH behavior. Axes are rescaled (by a factor of 8) to match RG predictions.

For Hermitian operators $\vec{A}$ and $\vec{B}$, the term $i\vec{J} \cdot (\vec{A} \times \vec{B})$ is Hermitian provided that the coupling vector $\vec{J}$ is real and the operators $\vec{A}$ and $\vec{B}$ commute. To see this, consider the Hermitian conjugate:

$$\left(i\vec{J} \cdot (\vec{A} \times \vec{B})\right)^{\dagger} = -i\vec{J} \cdot (\vec{B}^{\dagger} \times \vec{A}^{\dagger}),$$

where we used the properties $(\vec{A} \times \vec{B})^{\dagger} = \vec{B}^{\dagger} \times \vec{A}^{\dagger}$ and $\vec{J}^* = \vec{J}$. Since $\vec{A}$ and $\vec{B}$ are Hermitian and commute, we have

$$\vec{B}^{\dagger} \times \vec{A}^{\dagger} = \vec{B} \times \vec{A} = -\vec{A} \times \vec{B},$$

so that

$$\left(i\vec{J} \cdot (\vec{A} \times \vec{B})\right)^{\dagger} = i\vec{J} \cdot (\vec{A} \times \vec{B}),$$

confirming that the operator is indeed Hermitian under the stated conditions.

In the specific model under consideration, Hermiticity is preserved for momenta $k = k'$ and angular coordinates $\theta = \theta'$. However, more general inclusions such as potential scattering can introduce NH terms, as discussed in Sec. 5. Before turning to the full model, we show that even at the one-loop level, emergent non-Hermiticity appears. This arises from the RG invariant which will modify symmetry properties, which we analyze below. In order to discuss the pseudo-chiral symmetry and emergent non-Hermiticity of the effective model (12), we

write the local Hamiltonian at $k = 0$ in the spin basis as follows:

$$
\hat{\tilde{H}} = \tilde{J}_0\left(\sigma_z \otimes \boldsymbol{\sigma}_z + \sigma^+ \otimes \boldsymbol{\sigma}^- + \sigma^- \otimes \boldsymbol{\sigma}^+\right) + i\tilde{J}_{k^3}(\sigma_x \otimes \boldsymbol{\sigma}_y - \boldsymbol{\sigma}_y \otimes \sigma_x) \\
+ i\tilde{J}_k(\boldsymbol{\sigma}_y \otimes \boldsymbol{\sigma}_z - \boldsymbol{\sigma}_z \otimes \sigma_y) + i\tilde{J}_k(\sigma_x \otimes \boldsymbol{\sigma}_z - \boldsymbol{\sigma}_z \otimes \sigma_x).
\tag{13}
$$

We can construct the ket vector for the above Hamiltonian as $(|\uparrow\uparrow\rangle, |\uparrow\downarrow\rangle, |\downarrow\uparrow\rangle, |\downarrow\downarrow\rangle)$. For such a state, we can show the metric operator as the following:

$$
\eta = \begin{pmatrix} 0 & 0 & 0 & 1 \\ 0 & 0 & 1 & 0 \\ 0 & 1 & 0 & 0 \\ 1 & 0 & 0 & 0 \end{pmatrix}.
\tag{14}
$$

In the above EQ (13) the block symbol is for the bath spin and rest are impurity spin operators to distinguish between the two. The chiral symmetry operator can be generalized to $n$-dimensional matrices [21], is $\eta = \boldsymbol{\sigma}_x \otimes \sigma_x$. This metric satisfies $\eta\tilde{H}\eta^{-1} = -\tilde{H}^\dagger$, indicating pseudo-chiral symmetry [19] this inherent property become very crucial after adding the potential scattering terms which is eloborated in section 5. For Hermitian matrix this symmetry will be $\{\hat{H}, \eta\} = 0$. Each of the states can be represented as follows:

$$
\langle\uparrow\uparrow| = \begin{pmatrix} 1 & 0 & 1 & 0 \end{pmatrix}, \qquad \langle\uparrow\downarrow| = \begin{pmatrix} 1 & 0 & 0 & 1 \end{pmatrix}, \\
\langle\downarrow\uparrow| = \begin{pmatrix} 0 & 1 & 1 & 0 \end{pmatrix}, \qquad \langle\downarrow\downarrow| = \begin{pmatrix} 0 & 1 & 0 & 1 \end{pmatrix}.
\tag{15}
$$

So it can be readily seen the operations as $\eta|\uparrow\uparrow\rangle \to |\downarrow\downarrow\rangle$, $\eta|\uparrow\downarrow\rangle \to |\downarrow\uparrow\rangle$, $\eta|\uparrow\downarrow\rangle \to |\downarrow\uparrow\rangle$ and $\eta|\downarrow\downarrow\rangle \to |\uparrow\uparrow\rangle$.

$$
\hat{\tilde{H}} = \begin{pmatrix} \tilde{J}_0 & -\tilde{J}_k e^{-i\frac{\pi}{4}} & \tilde{J}_k e^{-i\frac{\pi}{4}} & 0 \\ -\tilde{J}_k e^{i\frac{\pi}{4}} & -\tilde{J}_0 & 2i\tilde{J}_{k^3} + \tilde{J}_0 & -\tilde{J}_k e^{-i\frac{\pi}{4}} \\ \tilde{J}_k e^{i\frac{\pi}{4}} & -2i\tilde{J}_{k^3} + \tilde{J}_0 & -\tilde{J}_0 & \tilde{J}_k e^{-i\frac{\pi}{4}} \\ 0 & -\tilde{J}_k e^{i\frac{\pi}{4}} & \tilde{J}_k e^{i\frac{\pi}{4}} & \tilde{J}_0 \end{pmatrix}.
\tag{16}
$$

The above matrix has the properties due to the ADM interaction,

$$
\eta_{\sigma_z \otimes \sigma_z} \hat{H} \eta_{\sigma_z \otimes \sigma_z}^{-1} = H, \quad \text{for} \quad J_k \to -J_k, \\
\eta_{\sigma_y \otimes \sigma_y} \hat{H} \eta_{\sigma_y \otimes \sigma_y}^{-1} = H, \quad \text{for} \quad J_{k^3} \to 0.
\tag{17}
$$

Choosing the positive invariant $(\tilde{J}_k \to J_{k+} = \frac{1}{2}(1 + \sqrt{1 - 4mJ_{k^3}}))$ and for either $m, J_{k^3} < 0$ and $J_0, J_{k^3}$ are same as running couplings to bare ones. We rewrite the EQ (16) as following,

$$
\hat{\tilde{H}}_{flown} = \begin{pmatrix} J_0 & -J_{k+} e^{-i\frac{\pi}{4}} & J_{k+} e^{-i\frac{\pi}{4}} & 0 \\ -J_{k+} e^{i\frac{\pi}{4}} & -J_0 & 2iJ_{k^3} + J_0 & -J_{k+} e^{-i\frac{\pi}{4}} \\ J_{k+} e^{i\frac{\pi}{4}} & -2iJ_{k^3} + J_0 & -J_0 & J_{k+} e^{-i\frac{\pi}{4}} \\ 0 & -J_{k+} e^{i\frac{\pi}{4}} & J_{k+} e^{i\frac{\pi}{4}} & J_0 \end{pmatrix}.
\tag{18}
$$

The above model (18) we can split into $\hat{H}_{flown}^{4mJ_{k^3}<1}$ and $\hat{H}_{flown}^{4mJ_{k^3}>1}$ so in the regime $4mJ_{k^3} > 1$ the coupling $J_{k\pm} = \frac{1}{2}(1 \pm i\sqrt{4mJ_{k^3}-1}) = \sqrt{mJ_{k^3}}e^{\pm i\chi}$ where the $\chi = \arctan\left(\sqrt{4mJ_{k^3}-1}\right)$. So it can be shown easily $\left(\hat{H}_{flown}^{4mJ_{k^3}<1}\right)^\dagger = \hat{H}_{flown}^{4mJ_{k^3}<1}$ but the other regime model can be simplified as following,

$$
\hat{H}_{flown}^{4mJ_{k^3}>1} = \begin{pmatrix} J_0 & \sqrt{mJ_{k^3}}e^{-i(\chi+\frac{\pi}{4})} & \sqrt{mJ_{k^3}}e^{i(\chi-\frac{\pi}{4})} & 0 \\ \sqrt{mJ_{k^3}}e^{-i(\chi-\frac{\pi}{4})} & -J_0 & 2iJ_{k^3} + J_0 & \sqrt{mJ_{k^3}}e^{-i(\chi+\frac{\pi}{4})} \\ \sqrt{mJ_{k^3}}e^{i(\chi+\frac{\pi}{4})} & -2iJ_{k^3} + J_0 & -J_0 & \sqrt{mJ_{k^3}}e^{i(\chi-\frac{\pi}{4})} \\ 0 & \sqrt{mJ_{k^3}}e^{-i(\chi-\frac{\pi}{4})} & \sqrt{mJ_{k^3}}e^{i(\chi+\frac{\pi}{4})} & J_0 \end{pmatrix}.
\tag{19}
$$

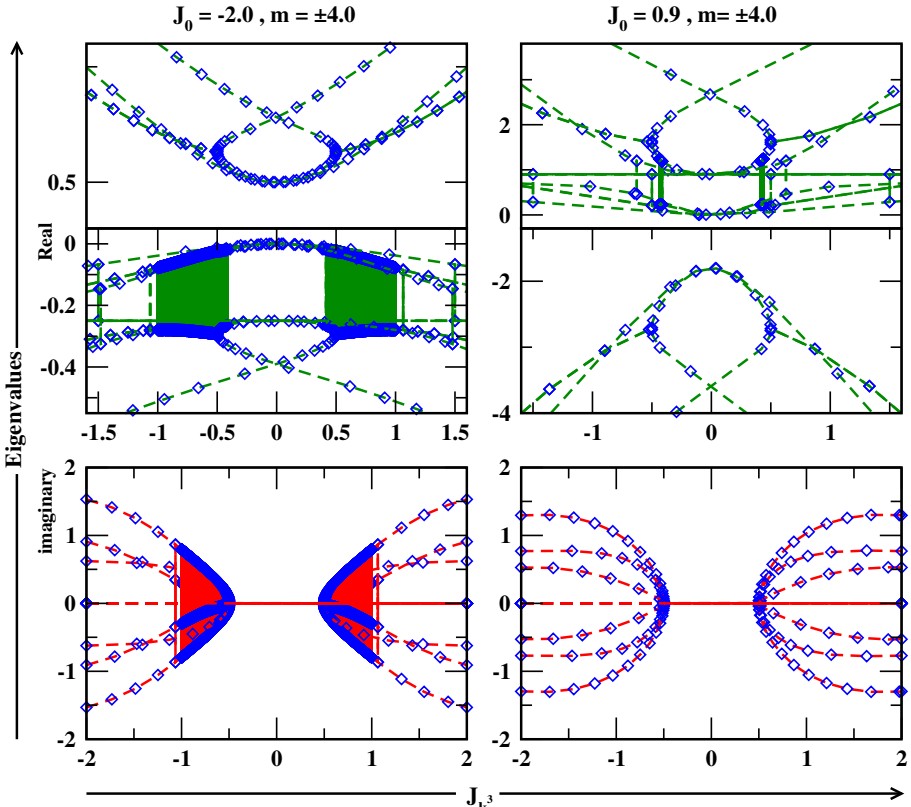

Figure 3: Eigenvalue spectra of the NH model (16) versus $J_{k^3}$ for $J_0 = -2.0$ (left) and 0.9 (right) with RG invariant $m = \pm 4.0$. Top and middle rows show real parts of eigenvalues $E_{1-4}$; bottom row shows imaginary parts. Dashed lines are RG analytical results; shaded regions mark spectral broadening and exceptional points (EPs) near $J_{k^3} \approx \pm 0.5, 0.75$, evidencing dissipative topology.

We refer to the regime $H_{\text{flown}}^{4mJ_{k^3}>1}$ as the **emergent NH model**, whose symmetry properties are discussed as follows.

$$\eta_{\sigma_x \otimes \sigma_x} H_{\text{flown}}^{4mJ_{k^3}>1} \eta_{\sigma_x \otimes \sigma_x}^{-1} = \left( H_{\text{flown}}^{4mJ_{k^3}>1} \right)^{\dagger}, \tag{20}$$

and, in the limit $J_0 \to 0$, exhibits a pseudo-chiral symmetry of the form

$$\eta_{\sigma_z \otimes \sigma_z} H_{\text{flown}}^{4mJ_{k^3}>1} \eta_{\sigma_z \otimes \sigma_z}^{-1} = -\left( H_{\text{flown}}^{4mJ_{k^3}>1} \right). \tag{21}$$

We analyze the eigenvalues of the spin Hamiltonian in the $(1, 1)$ occupancy sector, noting that complex eigenvalues and exceptional points (EPs) arise from perturbative renormalization group (RG) flow, influenced by the RG-invariant parameter $\tilde{J}_k$.

In Fock-space diagonalization for the single-occupancy sector(SS), we find that for $|J_{k^3}| < 0.5$, eigenvalues follow conventional Kondo regime with no EP's and topological transitions. Beyond this threshold, topological transitions occur, especially at high anisotropy $m$, as illustrated in Figures 2 and 3.

To compare with RG-derived fixed point theory (see Figure 7 second row ), we rescale the axes in our figures 2 and 3 by a factor of 8. The fixed points indicate real phase transitions, revealing a complex topological structure in the NH impurity model.

As $J_0$ increases, the Dirac cone collapses, resulting in a gapped spectrum and emphasizing the significance of $\tilde{J}_k$. EPs may fade as $J_0 \to \infty$, leaving strong coupling behavior an open question.

## 4.1 Condition number in Fock space

The condition number [22] $\kappa(A)$ of a matrix $A$ is defined as $\kappa(A) = \frac{|\lambda_{\max}|}{|\lambda_{\min}|}$, where $\lambda_{\max}$ and $\lambda_{\min}$ are the eigenvalues of largest and smallest magnitude, respectively. This quantity characterizes numerical sensitivity, the potential for error amplification, and the singular value spectrum of both real and complex matrices.

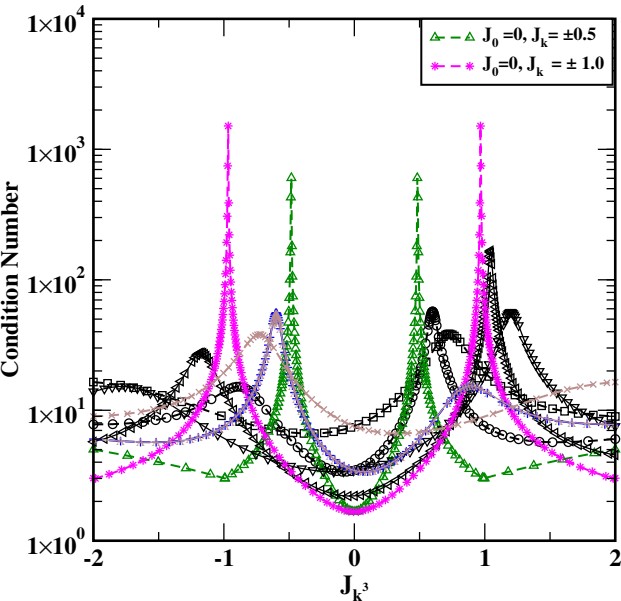

Figure 4: Condition number as a function of the ADM coupling $J_{k^3}$ in the model of EQ (13), computed using Fock-space diagonalization in the SS of the one-impurity problem. Diverging cusp-like features indicate large condition numbers, correlating with defects in diagonalization visible in Figure 3.

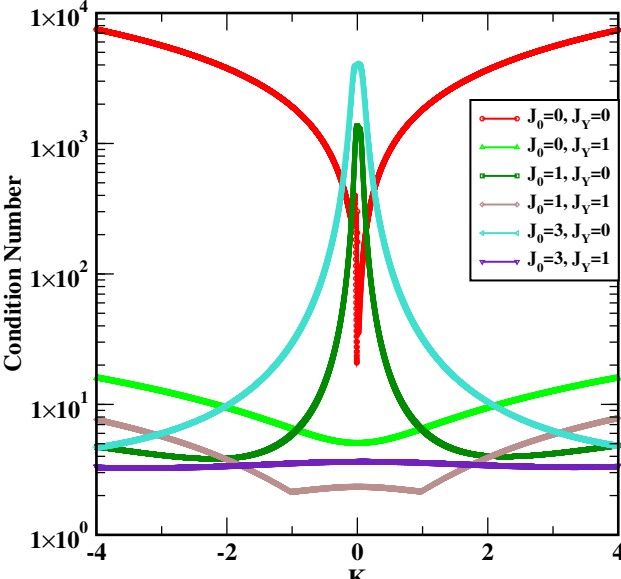

Figure 5: Variation of the condition number in Fock space for the two-impurity model defined in EQ (27). The model remains well-conditioned for most values of $K$, except near $K = 0$. Notably, large condition numbers emerge when ADM couplings are present, particularly for nonzero $J_{k^3}$, indicating enhanced sensitivity. In contrast, the model appears non-defective when both $J_{k^3}$ and $J_k$ are absent.

In Figures 4 and 5, we plot the condition number for the one- and two-impurity local spin Hamiltonians in Fock space. These plots reveal how the sensitivity of the system varies with the ADM-type interactions, namely the $J_{k^3}$ and $K$ couplings. A high condition number signals increased sensitivity and, in particular, the onset of near-defectiveness in diagonalization due to eigenvectors becoming nearly parallel—i.e., a loss of orthogonality—characteristic of exceptional points (EPs). We emphasize that these conclusions are drawn from the local Hamiltonian restricted to the SS of Fock space. Therefore, they pertain specifically to dissipation and NH effects in this reduced subsystem and may not directly extend to the full many-body Hilbert space. Nevertheless, dissipation in large-$N$ systems is generally understood to be governed by local subsystems, as global equilibration tends to wash out nonlocal features. Recently, the use of condition number analysis has been proposed as a diagnostic tool to probe perturbative sensitivity in NH systems [10], further motivating our investigation here.

## 5 Renormalization with potential scattering

We investigate the role of potential scattering in the renormalization of the effective impurity model, focusing on the fate of the RG-invariant responsible for eigenvalue coalescence—namely, the emergence of exceptional points (EPs)—in the presence of NH perturbations. Our central question is whether this invariant remains robust under renormalization group (RG) flow or acquires corrections at higher-loop orders, particularly at third order.

To address this, we analyze the renormalized couplings in the effective model, introducing a regular hexagonal bath geometry (Fig. 6) that naturally accommodates $\mathcal{PC}$-symmetric potential scattering terms. The structure of spin-spin interactions in this setting parallels the model in Ref. [8]. Notably, complex-valued potential scattering terms are essential for realizing RG fixed points, which signal topological transitions in the eigenvalue spectrum of the effective Hamiltonian. These spectral transitions correspond to qualitative changes in the ground state and are inherently tied to the NH nature of the model.

The effective model includes both ADM spin exchange and complex-valued scattering processes. We explore is whether the invariant associated with EPs is preserved under RG flow—implying topological robustness—or modified due to NH potential scattering, indicating an RG-generated origin of EPs.

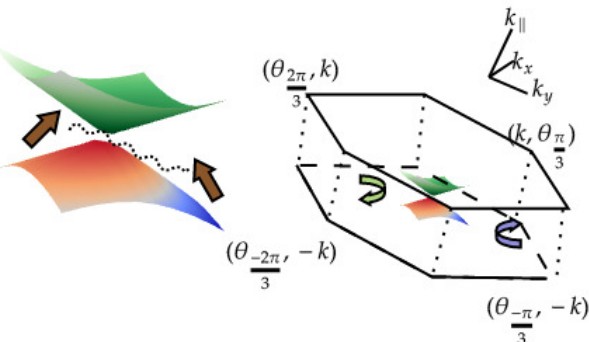

Figure 6: Left: Two Anderson impurities coupled to edge states of a Dirac-like bath. Right: Regular hexagonal bath geometry designed to incorporate $\mathcal{PC}$-symmetric potential scattering terms. Green and blue arrows denote left and right scattering centers, respectively. Variables $k_{\parallel} \to k$, $\sqrt{k_x^2 + k_y^2} \to k$, and $\theta \to \tan^{-1}(k_y/k_x)$. See Eqs. (3)–(5) for derivations. Electron and hole scattering from $(k, \theta)$ to $(-k, \theta')$ are depicted around the Dirac cone. The angular direction of momentum determines the sign of particle/hole states.

The effective Hamiltonian is given by:

$$H'_{\text{eff}} = H_0 + \sum_{kk'} J_0 \vec{s} \cdot \psi^\dagger \boldsymbol{\sigma} \psi + i \sum_{kk'} \vec{J}_{k^3}^{0,0} \cdot (\vec{s} \times \psi^\dagger \boldsymbol{\sigma} \psi) + i \sum_{kk'} \vec{J}_k^{(0,0)} \cdot (\vec{s} \times \psi^\dagger \boldsymbol{\sigma} \psi) + H_{\text{pot}}, \quad (22)$$

with the $\mathcal{PC}$-symmetric potential scattering term $H_{\text{pot}}$ defined as:

$$H_{\text{pot}}^{\text{NH}} = \sum_{kk'} \left( \vec{J}_{k^3}^{\frac{2\pi}{3},-\frac{\pi}{3}} + \vec{J}_k^{\frac{2\pi}{3},-\frac{\pi}{3}} \right) \cdot (\vec{s} \times \psi^\dagger \boldsymbol{\Gamma} \psi) - \sum_{kk'} \left( \vec{J}_{k^3}^{\pm\frac{2\pi}{3},\pm\frac{\pi}{3}} + \vec{J}_k^{\pm\frac{2\pi}{3},\pm\frac{\pi}{3}} \right) \cdot (\vec{s} \times \psi^\dagger \boldsymbol{\Omega} \psi). \quad (23)$$

To accommodate the structure of the NH Kondo model, we define momentum-resolved creation operators as:

$$\psi^\dagger = \left( c_{k,\frac{2\pi}{3},+}^\dagger, \quad c_{k,\frac{\pi}{3},-}^\dagger, \quad c_{-k,-\frac{2\pi}{3},+}^\dagger, \quad c_{-k,-\frac{\pi}{3},-}^\dagger \right),$$

enabling a block-structured representation of the impurity-bath coupling via generalized Pauli matrices. The RG EQs derived from this effective Hamiltonian are detailed in Appendix B, based on a third-order perturbative expansion. These form the basis for the flow analysis shown in Fig. 7.

For compactness, we relabel the couplings as:

$$\left( J_0, J_{k^3}^{0,0}, J_{k^3}^{\pm\frac{2\pi}{3},\pm\frac{\pi}{3}}, J_{k^3}^{\frac{2\pi}{3},-\frac{\pi}{3}}, J_k^{(0,0)}, J_k^{\pm\frac{2\pi}{3},\pm\frac{\pi}{3}}, J_k^{\frac{2\pi}{3},-\frac{\pi}{3}} \right) \rightarrow \left( J_0, J_{k^3}, g_{1k^3}, g_{2k^3}, J_k, g_{1k}, g_{2k} \right).$$

Among the couplings $\{g_{1k^3}, g_{2k^3}, g_{1k}, g_{2k}\}$, which originate from potential scattering terms, the invariant derived at one-loop remains unrenormalized. However, these terms generate additional invariants, giving rise to symmetric spiral-points (SP) in the RG flow diagrams (Fig. 7). This reveals a topologically nontrivial interplay between NH scattering and RG evolution, mediated by NL couplings.

# 6 Generalization to two impurities

We extend our model to a two-impurity Kondo system, incorporating direct spin-spin coupling and an isotropic DM interaction between impurities. Similar interactions have been studied in double quantum dots with spin-orbit coupling [23], where the DM term is restricted to the Y-component for baths with linear dispersion. Here, we analyze the renormalization of ADM versus isotropic interactions.

## 6.1 Effective Hamiltonian and renormalization group equations

The effective spin Hamiltonian takes the form:

$$H_{\text{eff}} = H_0 + \sum_{kk'} J_0 s_\alpha \cdot \mathbf{S}_{kk'} + i \sum_{kk'\zeta} \vec{J}_{k^3} \cdot (s_\alpha \times \mathbf{S}_{kk'}) + i \sum_{kk'\zeta} \vec{J}_k \cdot (s_\alpha \times \mathbf{S}_{kk'}) + J_Y s_1 \cdot s_2 + i\vec{K} \cdot (s_1 \times s_2), \tag{24}$$

where $H_0 = \sum_{k\zeta} \epsilon_{k\zeta} c_{k\zeta}^\dagger c_{k\zeta}$ is the kinetic energy term, $J_Y$ represents the direct coupling between impurities, and $\vec{K}$ denotes the impurity DM interaction. The bold symbols denote spinor vectors, as previously defined Since the generalized problem introduces multiple couplings, we

restrict our analysis to the one-loop RG EQs:

$$\frac{dJ_0}{dl} = J_0^2 + J_{k^3}J_k + J_{k^3}^2 + J_k^2 + J_Y J_0 + K J_0,$$

$$\frac{dJ_{k^3}}{dl} = J_k^2 + J_0 J_{k^3} + J_Y J_{k^3} + J_Y J_k + K J_{k^3},$$

$$\frac{dJ_k}{dl} = J_0 J_k + J_Y J_{k^3} + J_Y J_k + K J_k, \tag{25}$$

$$\frac{dJ_Y}{dl} = J_Y^2 + K^2 + J_0^2 + J_{k^3}J_k + J_k^2 + J_{k^3}^2,$$

$$\frac{dK}{dl} = K^2 + K J_Y + J_{k^3}J_k.$$

Figure 7: RG flow diagram for the one-impurity model with fixed invariant values $g_{1k} = g_{1k^3} = 1$, derived in Appendix B. Couplings are parametrized as $J_k = \frac{1}{2}\left(1 \pm \sqrt{1 + 4mJ_{k^3}}\right)$ with $m = -4$. Top row: Flow in the potential scattering channels. Second and third rows: Complex RG flows of $J_k$, showing fixed points (FPs), spiral points (SPs), and a marginal point. Dotted lines indicate a family of fixed points. Signatures of exceptional points appear in the lower-left quadrant, where nearly parallel flow lines emerge for small $J_0$ and $J_{k^3}$, indicating unstable directions. These SPs signal topological spectral transitions (see Fig. 2) and correspond to large condition numbers (Fig. 4), especially near $J_{k^3} = 0.5$.

Solutions to the RG EQs in various limits are discussed in Appendix B.1. The vanishing of beta functions, signaling Kondo destruction, is analyzed within the odd-even impurity framework of Refs. [24, 25]. Extending this, we explore how ADM and NL couplings $J_k$ and $J_{k^3}$ lead to anomalous spin relaxation and novel fixed points beyond the isotropic regime.

## 6.2 Eigenspectrum and fixed points

To further understand the connection between RG fixed points and the eigenspectrum, we construct the three-spin Fock space representation for the two-impurity system which is represented schematically in 8. The effective Hamiltonian in this basis is:

$$\tilde{H}_{2\text{imp}} = \sum_{kk'} \tilde{J}_0 s_\alpha \cdot \mathbf{S}_{kk'} + i \sum_{kk'\alpha} \tilde{\vec{J}}_{k^3} \cdot (s_\alpha \times \mathbf{S}_{kk'}) + i \sum_{kk'\alpha} \tilde{\vec{J}}_k \cdot (s_\alpha \times \mathbf{S}_{kk'}) + \tilde{J}_Y s_1 \cdot s_2 + i\tilde{\vec{K}} \cdot (s_1 \times s_2). \quad (26)$$

Here, $\alpha$ indexes the two impurities. The Fock space representation in the symmetric sector (SS) is given by:

$$\psi^\dagger = (|\uparrow\uparrow\uparrow\rangle, |\uparrow\uparrow\downarrow\rangle, |\uparrow\downarrow\uparrow\rangle, |\downarrow\uparrow\uparrow\rangle, |\uparrow\downarrow\downarrow\rangle, |\downarrow\downarrow\uparrow\rangle, |\downarrow\uparrow\downarrow\rangle, |\downarrow\downarrow\downarrow\rangle).$$

The Hamiltonian is then expressed as:

$$\hat{\tilde{H}}_{2\text{imp}} = \langle \psi | \tilde{H}_{2\text{imp}} | \psi \rangle. \quad (27)$$

Expanding in matrix form, we analyze the symmetry properties of $\hat{\tilde{H}}_{2\text{imp}}$. Constructing the metric operator $\hat{\eta} = \sigma_x \otimes \sigma_x \otimes \sigma_x$ as in Ref. [21], we verify its $\mathcal{PC}$ symmetry, expressed as:

$$\hat{\eta}\hat{H}\hat{\eta}^\dagger = -\hat{H}^\dagger.$$

We show that ADM-interaction is necessary to satisfy this condition, specifically requiring $J_{k^3} \neq 0$. The Hamiltonian lacks $\mathcal{PC}$ symmetry when $J_{k^3} = 0$. The full Hamiltonian matrix is given by: This matrix can be compactly expressed in a $4 \times 4$ block matrix form as:

$$\hat{\tilde{H}}_{2imp} = \begin{pmatrix} \mathbb{M}_1 & \mathbb{M}_2 \\ \mathbb{M}_3 & \mathbb{M}_4 \end{pmatrix},$$

where $\mathbb{M}_1$, $\mathbb{M}_4$ are $4 \times 4$ diagonal block matrices, and $\mathbb{M}_2$, $\mathbb{M}_3$ contain off-diagonal coupling terms.

$$\mathbb{M}_1 = \begin{pmatrix} 2J_0 + J_Y & -e^{i\frac{\pi}{4}}(J_k + K) & 0 & 0 \\ -e^{-i\frac{\pi}{4}}(J_k + K) & -J_Y & 4(J_0 + K) & -2e^{i\frac{\pi}{4}}J_k \\ 0 & 4(J_0 - K) & -2J_0 + J_Y & e^{i\frac{\pi}{4}}(J_k - K) \\ 0 & -2e^{-i\frac{\pi}{4}}J_k & e^{-i\frac{\pi}{4}}(J_k - K) & -J_Y \end{pmatrix},$$

$$\mathbb{M}_2 = \begin{pmatrix} e^{i\frac{\pi}{4}}(J_k + K) & 0 & 0 & 0 \\ 4(J_Y + K) & e^{i\frac{\pi}{4}}(J_k - K) & 0 & 0 \\ 4(J_0 + K) & 0 & -e^{i\frac{\pi}{4}}(J_k - K) & 0 \\ 0 & 4(J_0 + K) & 4(J_Y + K) & -e^{i\frac{\pi}{4}}(J_k + K) \end{pmatrix},$$

$$\mathbb{M}_3 = \mathbb{M}_2^\dagger = \begin{pmatrix} e^{-i\frac{\pi}{4}}(J_k + K) & 4(J_Y + K) & 4(J_0 + K) & 0 \\ 0 & e^{-i\frac{\pi}{4}}(J_k - K) & 0 & 4(J_0 + K) \\ 0 & 0 & -e^{-i\frac{\pi}{4}}(J_k - K) & 4(J_Y + K) \\ 0 & 0 & 0 & -e^{-i\frac{\pi}{4}}(J_k + K) \end{pmatrix},$$

$$\mathbb{M}_4 = \begin{pmatrix} -J_Y & -e^{i\frac{\pi}{4}}(J_k - K) & -2e^{i\frac{\pi}{4}}J_k & 0 \\ -e^{-i\frac{\pi}{4}}(J_k - K) & -2J_0 + J_Y & 4(J_0 + K) & 0 \\ -2e^{-i\frac{\pi}{4}}J_k & 4(J_0 - K) & -J_Y & e^{i\frac{\pi}{4}}(J_k + K) \\ 0 & 0 & e^{-i\frac{\pi}{4}}(J_k + K) & 2J_0 + J_Y \end{pmatrix}.$$

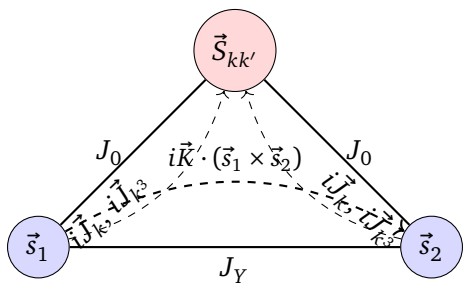

Figure 8: Schematic representation of the two-impurity Kondo model. The couplings $(J_0, J_Y)$ are shown as solid lines; DM couplings $(J_k, J_{k^3}, K)$ are shown as dashed arrows. The non-Hermiticity only enter through either substitution of invariant or through the potential scatterings.

**Symmetry properties:**

- **Diagonal block symmetry:** $\mathbb{M}_4$ is related to $\mathbb{M}_1$ via index-reversal, complex conjugation, and a sign alternation:
$$(\mathbb{M}_4)_{ij} = (-1)^{i-j} (\mathbb{M}_1)^*_{(5-j)(5-i)}.$$

  For example, $(\mathbb{M}_1)_{12} = -e^{i\pi/4}(J_k + K)$ corresponds to $(\mathbb{M}_4)_{43} = e^{-i\pi/4}(J_k + K)$.

- **Off-diagonal block symmetry:** $\mathbb{M}_3$ is the Hermitian conjugate of $\mathbb{M}_2$:
$$\mathbb{M}_3 = \mathbb{M}_2^\dagger.$$

- **Global pseudo-Hermiticity:** Let $P$ be the $4 \times 4$ anti-diagonal permutation matrix, i.e., $P_{ij} = \delta_{i,5-j}$. Define $D = \mathrm{diag}(1,-1,1,-1)$. Then:
$$\mathbb{M}_4 = DP\mathbb{M}_1^* P^{-1} D^{-1} \quad \Rightarrow \quad \hat{\hat{H}} = (\mathbb{1}_2 \otimes PD)\hat{\hat{H}}^*(\mathbb{1}_2 \otimes PD)^{-1}.$$

With substitution of RG-invariant it follows the symmetry discussions for NH regimes as in EQs (20) and (21). This matrix condition number is shown in figure 5. We show numerically $J_{k^3} \neq 0$ to exhibit large condition number in contrast to isotropic impurity DM interactions.

# 7 Numerical solutions and dissipative fixed points in RG flow

We solve the coupled renormalization group (RG) EQs in Eq. (25) numerically for a range of initial conditions near analytically identified fixed points. Figure 9 displays representative flow trajectories of the couplings, restricted to domains with real value. The evolution of $J_{k^3}$ and $J_k$ exhibits invariance consistent with our analytical predictions. We observe that the flow of couplings when $J_{k^3} \neq 0$ show extreme sensitivity. Presence AFM-divergences in cases $J_{k^3} \approx 0$ show the absence of dissipative corrections and usual Kondo-regime.

*Sign Reversion and Complex RG Analysis*:To further understand the structure of the flow near dissipative fixed points, we extend the RG EQ's into the complex plane by treating the couplings as complex variables. The resulting beta functions exhibit nontrivial analytic structure, including poles on imaginary axes, branch-like behavior, and spiral trajectories in the complex domain—features characteristic of NH or dissipative systems [26].

We identify this dissipative behavior through—*sign reversion* (SR)—in the imaginary part of the couplings, where the sign of the imaginary component undergoes an abrupt switch near

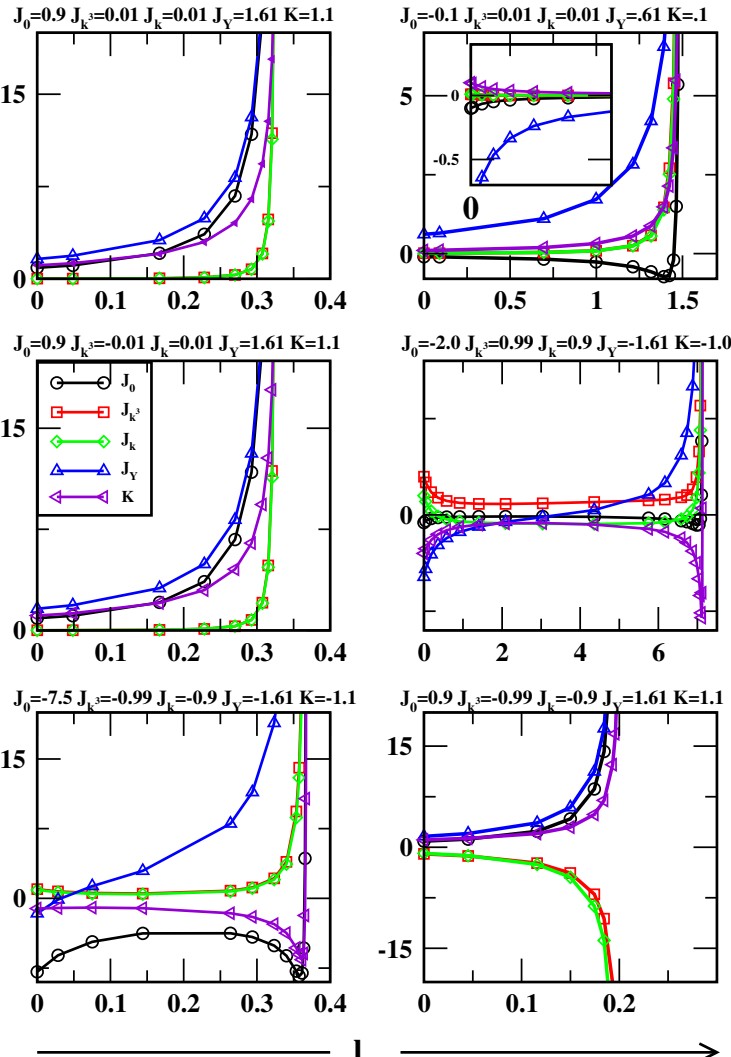

Figure 9: Numerical solutions of the RG flow EQs (25) near various fixed points. The horizontal axis corresponds to the RG flow parameter. Initial conditions for each trajectory are labeled. Diverging flows signal strong-coupling phases, whereas cusp-like sign-reversal features indicate dissipative fixed points. The inset shows flow of RKKY coupling to zero when all bath originated couplings are zer.

certain fixed points. These SR points coincide with what we term *spiral fixed points*(SPs), as illustrated in Figure 10. This signature behavior persists across both one- and two-impurity extensions of the model and serves as a diagnostic for dissipative or $\mathcal{PT}$-symmetric breaking transitions.

These results reveal a qualitative distinction between *dissipative spiral fixed points*, which show SR and complex-valued beta function structures, and *unstable fixed points*, where such features are absent. The existence of SR behavior offers a new diagnostic tool for classifying fixed points in open quantum systems with ADM, pseudochiral, or $\mathcal{PT}$-symmetric interactions.

*Consistency with Conformal Field Theory and RG Approaches*:The RG flows we present align with expectations from conformal field theory (CFT), particularly in the emergence of nontrivial topology and fixed-point classification in dissipative regimes. Our numerical results support the analytical beta function structure and confirm that RG and CFT provide complementary but consistent insights into the infrared behavior of the system. In particular, the identification of SR features and spiral flows establishes new phenomenology beyond traditional Hermitian RG fixed points.

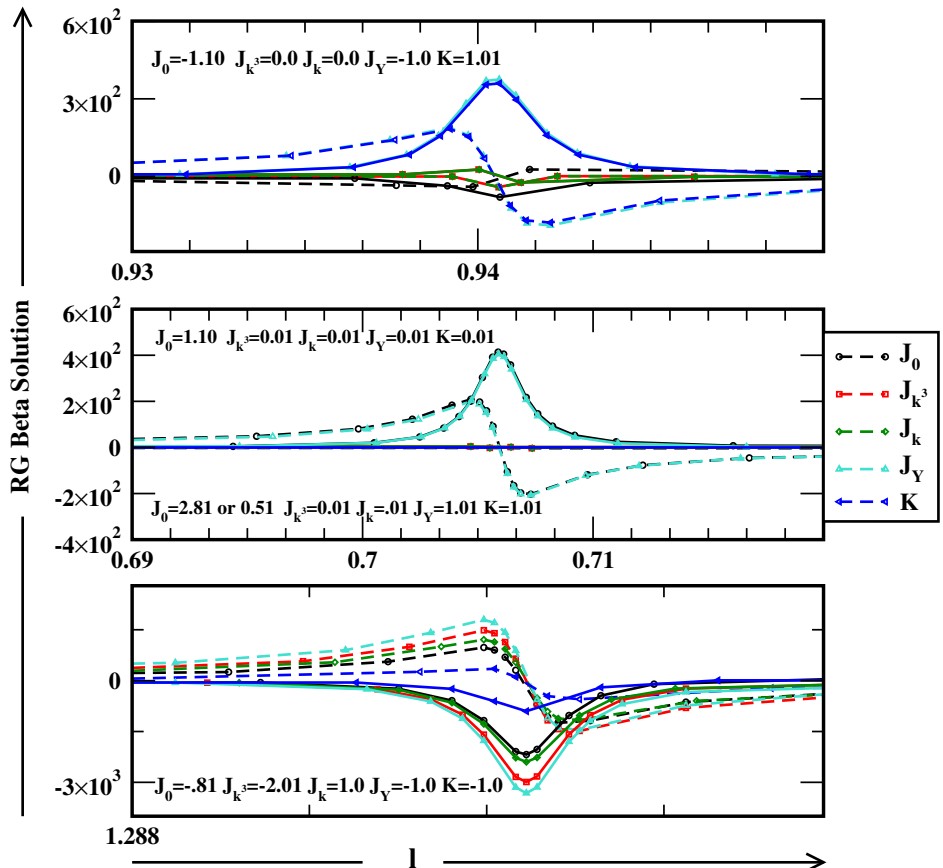

Figure 10: Flow of RG couplings in the complex domain. Dotted and solid lines represent real and imaginary components, respectively. In the lower panel, sign reversion in the imaginary part is observed when $J_{k^3} = -2.0$, a hallmark of spiral fixed points. The middle panel compares two different initial conditions. Between SR phases, multi-pole structures appear, hinting at rich underlying topology.

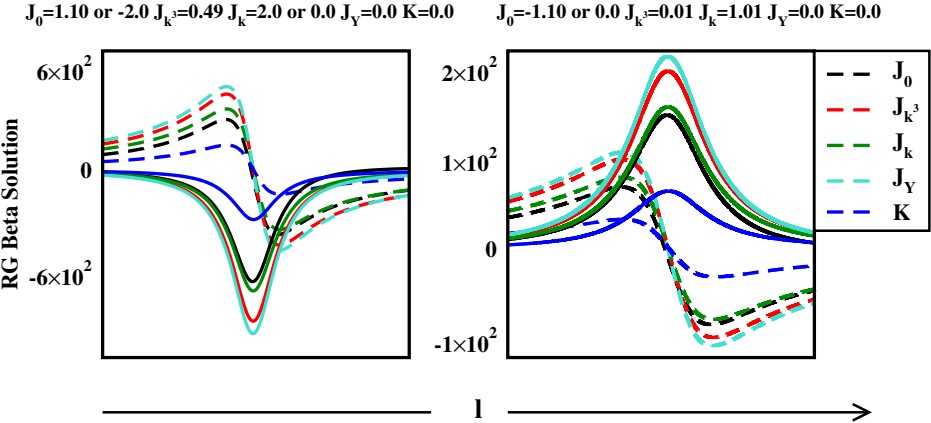

Figure 11: Sign reversion observed near specific fixed points (highlighted in red in Fig. 7, second row), contrasted with the absence of such behavior at an unstable point (in 7 bottom row black point). Small initial values for all other couplings ensure the regime corresponds to a single impurity.

# 8 Impurity transport calculation

We compute the anomalous contributions to the relaxation time in the presence of a NL dispersive bath using the $\mathcal{T}_{kk'}$ formalism [20], with detailed calculations provided in Appendix C. The expression for the relaxation time is given by:

$$\frac{1}{\tau} \propto (1 - 2J\tilde{g}_\alpha - 2J_{k^3}\tilde{g}_{\alpha k^3} - 2J_k\tilde{g}_{\alpha k}), \tag{28}$$

where the couplings $J_0, J_k, J_{k^3}$ correspond to the bare model parameters, while the integrals $\tilde{g}_\alpha, \tilde{g}_{\alpha k^3}$, and $\tilde{g}_{\alpha k}$ arise from perturbative RG and inclusion of impurity scattering (see Appendix C).

We further investigate the criticality associated with dissipative fixed points obtained from perturbative RG. Our analysis reveals two distinct scaling collapse regimes, as discussed in the next section, and an emergent invariant structure that persists in the RG flow.

## 8.1 Scaling collapse in $\frac{1}{\tau}$

In the preceding sections, we identified emergent coalescing points arising when the integrand in Eq. (C.8) vanishes or when $q \to 0$. These correspond to the condition

$$(k^2 - \mu)^2 - \lambda^2 k^2 = 0, \tag{29}$$

which yields momentum-space roots

$$\frac{k}{\lambda} = \frac{1}{2} \pm \frac{1}{2}\sqrt{1 + \frac{4\mu}{\lambda^2}}. \tag{30}$$

We compare this to the RG-invariant relation

$$J_k = \frac{1}{2} \pm \frac{1}{2}\sqrt{1 + 4mJ_{k^3}}, \tag{31}$$

and note that in the small-$\beta$ regime, the scalings $J_k \propto \frac{k}{\lambda}$ and $J_{k^3} \propto \frac{1}{\lambda^2}$ hold, with $\mu$ playing the role of the invariant $m$. This correspondence mirrors the equivalence encountered in laser-induced Kondo effect [27] where non-equilibrium steady state and poorman scaling is compared. Exceptional points (EPs) emerge only when the chemical potential $\mu \neq 0$, defining a critical regime where real-valued null momentum solutions exist. When $\mu$ exceeds a threshold, the momentum roots become complex, signaling the onset of a nontrivial gauge structure. These null momentum points also appear in the spectrum of the effective spin-spin Hamiltonian.

To visualize these effects, we plot the relaxation rate $\tau^{-1}$ in Fig. 12, fixing $\mu = -1.0$ and $\lambda = 1.0$, while varying the excitation energy $\epsilon$. The data is rescaled using the bandwidth and the momentum-root prescription described above to demonstrate scaling collapse. While a full analysis at strong dissipation with complex couplings is beyond the scope of this work, our results clearly establish the emergence of invariant structures under bath-induced perturbations.

In the ADM model, $\mathcal{PC}$-symmetric scattering gives rise to a NH Kondo effect. Under renormalization, special points (SPs) emerge, characterized by complex flow trajectories and possible topological transitions. Although additional scattering channels were omitted from the transport calculation, their impact may be addressed in future work using more refined techniques for dissipative systems.



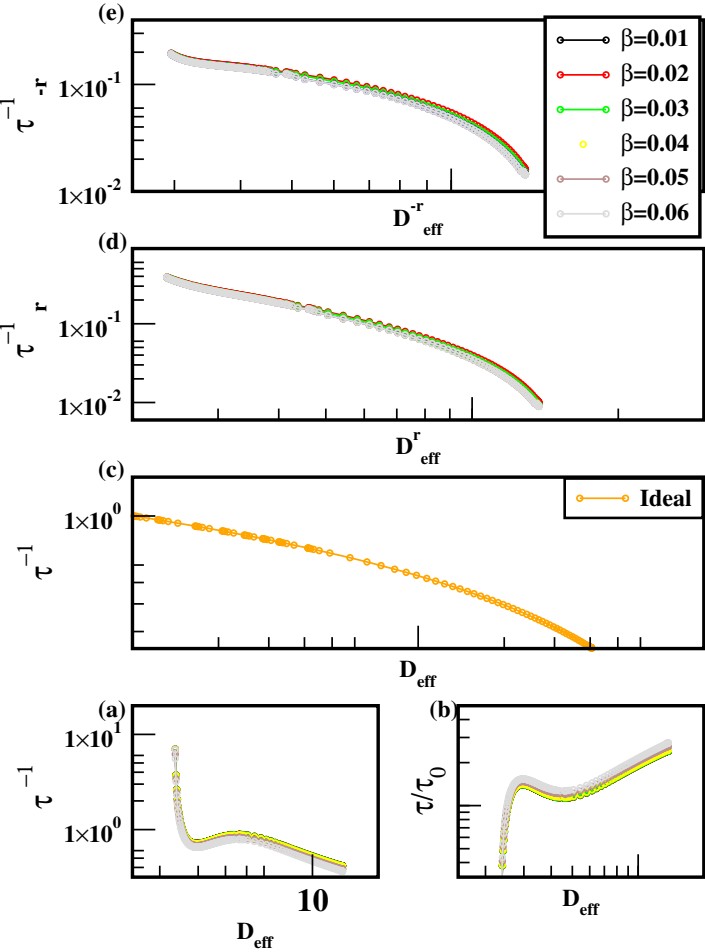

Figure 12: (a) Relaxation rate $\tau^{-1}$ as a function of excitation energy for various values of the NL parameter $\beta$; $\tau^{-1}$ is proportional to resistivity. (b) Normalized curves exhibit a kink structure at low energy. (c), (d) Scaling collapse of the curves using two distinct momentum-root prescriptions: $\tau_{\pm r}^{-1} = \tau^{-1} \times (1 \pm \sqrt{4D_{\text{eff}} + 1})$ and $D_{\text{eff}}^{\pm r} = D_{\text{eff}} \times (1 \pm \sqrt{4D_{\text{eff}} + 1})$. The kink vanishes upon rescaling, indicating emergent scale invariance.

## 9  Results and discussion

We have demonstrated the emergence of *exceptional points* (EPs) and nontrivial *renormalization group* (RG) behavior in a dissipative quantum impurity model, where $\mathcal{PC}$-symmetric potential scattering modifies the RG flow and fixed-point structure. Unlike in conventional Kondo models, where such perturbations are often RG-irrelevant under particle-hole symmetry, here they play a decisive role in the weak-to-intermediate coupling regime, signaling a departure from standard Kondo scaling.

A central result of our analysis is the presence of a shared *invariant structure* in both RG trajectories and spin-transport near dissipative fixed points. We further uncover *topological transitions* in the local Fock-space Hamiltonian that are closely aligned with the fixed points obtained from RG analysis. These transitions are accompanied by high condition numbers and indicate the onset of *defective eigenvalues*—a signature of NH degeneracies induced by anisotropy.

Our investigation of single- and two-impurity Kondo models in a homogeneous bath shows that ADM interactions act as Hermiticity-breaking yet $\mathcal{PC}$-symmetric perturbations. These

drive the system toward coalescing EPs in the RG flow. Furthermore, finite-momentum ($q$) bath modes introduce distinct transport features, while *sign-reversion* regimes near the fixed points reflect the role of potential scattering in rendering the system effectively open and non-Hermitian. NL perturbations, especially those with cubic momentum dependence, significantly deform the RG flow, giving rise to elliptic trajectories and emergent EPs that go beyond the linear paradigm.

These results suggest avenues for realizing *higher-order EPs* and related topological features in anisotropic spin-defect systems [28]. Potential experimental platforms include *diamond NV centers, cavity QED arrays* engineered systems and other systems where momentum-dependent DM interactions can be tuned. Our findings also motivate the development of *advanced numerical renormalization group (NRG)* techniques that incorporate complex-valued and ADM couplings—an extension not captured by standard NRG approaches.

In a forthcoming study [29], we employ Keldysh mean-field Green function techniques to explore non-equilibrium impurity dynamics. There, we find fixed points and invariant features similar to those reported here, including hybridization behavior near EPs, further reinforcing our present conclusions.

Finally, the mechanisms highlighted in this work—anisotropy, NL, and dissipation—are naturally realized in *two-dimensional thin films* and other low-dimensional systems, where *spin-orbit coupling* and structural asymmetry are prevalent. Experimental settings such as *quantum dot arrays* or materials with *vacancy-induced spin textures* may host effective NH impurity physics. Even *Floquet-engineered band structures* can generate cubic momentum couplings, offering additional pathways to explore this physics. While our results are primarily theoretical, the predicted EPs and their transport signatures may offer a valuable interpretive framework for experimental deviations from conventional Kondo behavior.

## Acknowledgments

We thank JNCASR for the supportive research environment. Special thanks to Prof. Henrik Johannesson for conceptualizing the TKSS model, and to Prof. Ipsita Mandal and Dr. Ranjani Seshadri for introducing Weyl and topological systems. VMK acknowledges discussions with Alexander Poddubny on the NH skin effect in a Hermitian model.

## A   Projection details for deriving effective sd model

The projection operator method is a versatile technique used in single-particle quantum mechanics [30] and many-body physics [20], particularly in the study of Kondo problems. It allows for the description of many-body wavefunctions in distinct sectors based on the occupancy of the impurity subspace, such as unoccupied ($\psi_0$), singly occupied ($\psi_1$), and doubly occupied ($\psi_2$) sectors. The method provides a formalism for projecting the total many-body wavefunction onto these sectors, facilitating a systematic analysis of the Kondo problem.

$$\left(\sum_{i=0}^{2}|\psi_i\rangle\langle\psi_i|\right)H\left(\sum_{i=0}^{2}|\psi_i\rangle\langle\psi_i|\right)\left(\sum_{i=0}^{2}|\psi_i\rangle\langle\psi_i|\right)\psi = E\left(\sum_{i=0}^{2}|\psi_i\rangle\langle\psi_i|\right)\psi. \qquad (A.1)$$

The identity operators can be represented as projectors onto the $(0,1,2)$ subspaces, corresponding to unoccupied, singly occupied, and doubly occupied states. These projectors are orthogonal.

$$
\begin{pmatrix} H_{00} & H_{01} & H_{02} \\ H_{10} & H_{11} & H_{12} \\ H_{20} & H_{21} & H_{22} \end{pmatrix} \begin{pmatrix} \psi_0 \\ \psi_1 \\ \psi_2 \end{pmatrix} = E \begin{pmatrix} \psi_0 \\ \psi_1 \\ \psi_2 \end{pmatrix}, \qquad \text{where} \qquad P_\eta P_{\eta'} = \delta_{\eta\eta'} P_{\eta=\eta'},
$$
$$
H_{\eta,\eta'=0,1,2} = P_\eta H P_{\eta'}, \qquad \text{and} \qquad P^I = P, \quad I \in \mathcal{Z}^+. \tag{A.2}
$$

We can define the projection operators using impurity number operators satisfying the completness relation $\sum_\eta P_\eta = \mathcal{I}$ as $P_0 = (1-n_\downarrow)(1-n_\uparrow), P_1 = n_\downarrow(1-n_\uparrow) + n_\uparrow(1-n_\downarrow)$, and $P_2 = n_\uparrow n_\downarrow$. The effective Hamiltonian for different subspaces can be obtained by eliminating the corresponding wave function in favor of the others.

$$
H_{eff}^\eta = H_{\eta\eta} + \sum_{\substack{\eta' \neq \eta = 0,1,2 \\ \zeta,\zeta' = \pm}} H_{\eta\eta'}^\zeta \frac{1}{E - H_{\eta'\eta'}} H_{\eta'\eta}^{\zeta'}. \tag{A.3}
$$

In the EQ (A.3), the index $\eta$ and $\eta' = 0,1,2$ corresponds to the unoccupied, singly occupied, and doubly occupied states, respectively. Computing the $P_1 H P_2$ we get the following,

$$
\tilde{H}_{12}^+ = \sum_{k\theta} \tilde{V}_k^\theta \alpha_{k1} e^{-i\frac{\theta}{2}} c_{k+}^\dagger n_\downarrow d_\uparrow + \sum_{k\theta} e^{i\frac{\theta}{2}} \tilde{V}_k^\theta \alpha_{k2} c_{k+}^\dagger n_\uparrow d_\downarrow,
$$
$$
\tilde{H}_{12}^- = \sum_{k\theta} i\tilde{V}_k^\theta \alpha_{k2} e^{-i\frac{\theta}{2}} c_{k-}^\dagger n_\downarrow d_\uparrow - \sum_{k\theta} i e^{i\frac{\theta}{2}} \tilde{V}_k^\theta \alpha_{k1} c_{k-}^\dagger n_\uparrow d_\downarrow \tag{A.4}
$$
$$
\implies \quad (H_{21}^+ + H_{21}^-) = (H_{12}^+ + H_{12}^-)^\dagger.
$$

Similarly for the $P_0 H P_1$ again only hybridization terms contribute can be shown as the following,

$$
\tilde{H}_{10}^+ = \sum_{k\theta} \tilde{V}_k^\theta \alpha_{k1} e^{-i\frac{\theta}{2}} c_{k+}^\dagger (1-n_\downarrow) d_\uparrow + \sum_{k\theta} e^{i\frac{\theta}{2}} \tilde{V}_k^\theta \alpha_{k2} c_{k+}^\dagger (1-n_\uparrow) d_\downarrow,
$$
$$
\tilde{H}_{10}^- = \sum_{k\theta} i\tilde{V}_k^\theta \alpha_{k2} e^{-i\frac{\theta}{2}} c_{k-}^\dagger (1-n_\downarrow) d_\uparrow - \sum_{k\theta} i e^{i\frac{\theta}{2}} \tilde{V}_k^\theta \alpha_{k1} c_{k-}^\dagger (1-n_\uparrow) d_\downarrow \tag{A.5}
$$
$$
\implies \quad (H_{01}^+ + H_{01}^-) = (H_{10}^+ + H_{10}^-)^\dagger.
$$

The components $H_{02}$ and $H_{20}$ will vanish since $P_0$ commutes and also using the orthogonality $P_0 P_2 = 0$. The remaining components can be computed as follows:

$$
H_{00} = \sum_{k\zeta} \epsilon_{k\zeta} c_{k\zeta}^\dagger c_{k\zeta} P_0 + \sum_\sigma \epsilon_d n_\sigma P_0,
$$
$$
H_{11} = \sum_{k\zeta} \epsilon_{k\zeta} c_{k\zeta}^\dagger c_{k\zeta} P_1 + \sum_\sigma \epsilon_d n_\sigma P_1, \tag{A.6}
$$
$$
H_{22} = \sum_{k\zeta} \epsilon_{k\zeta} c_{k\zeta}^\dagger c_{k\zeta} P_2 + \sum_\sigma \epsilon_d n_\sigma P_2 + U n_\uparrow n_\downarrow P_2.
$$

We will derive the components of the Hamiltonian to obtain the effective Hamiltonian can be expressed as:

$$
H_{eff}^1 = H_{11} + \sum_{\zeta,\zeta'=\pm} \left( H_{12}^\zeta \frac{1}{E - H_{22}} H_{21}^{\zeta'} + H_{10}^\zeta \frac{1}{E - H_{00}} H_{01}^{\zeta'} \right).
$$

The derivation proceeds as follows:

$$\sum_{\zeta,\zeta'=\pm} H_{12}^{\zeta}\frac{1}{E-H_{22}}H_{21}^{\zeta'} = H_{12}^{+}\frac{1}{E-H_{22}}H_{21}^{+} + H_{12}^{+}\frac{1}{E-H_{22}}H_{21}^{-} + H_{12}^{-}\frac{1}{E-H_{22}}H_{21}^{+} + H_{12}^{-}\frac{1}{E-H_{22}}H_{21}^{-}$$

$$= \sum_{kk'\theta\theta'} \tilde{V}_k^{\theta}\tilde{V}_{k'}^{*\theta'}\left(\alpha_{k1}e^{-i\frac{\theta}{2}}c_{k+}^{\dagger}n_{\downarrow}d_{\uparrow} + e^{i\frac{\theta}{2}}\alpha_{k2}c_{k+}^{\dagger}n_{\uparrow}d_{\downarrow}\right)$$

$$\times \hat{M}_{22}\left(\alpha_{k'1}e^{-i\frac{\theta'}{2}}c_{k'+}^{\dagger}n_{\downarrow}d_{\uparrow} + e^{i\frac{\theta'}{2}}\alpha_{k'2}c_{k'+}^{\dagger}n_{\uparrow}d_{\downarrow}\right)^{\dagger}$$

$$+ \sum_{kk'\theta\theta'} \tilde{V}_k^{\theta}\tilde{V}_{k'}^{*\theta'}\left(\alpha_{k1}e^{-i\frac{\theta}{2}}c_{k+}^{\dagger}n_{\downarrow}d_{\uparrow} + e^{i\frac{\theta}{2}}\alpha_{k2}c_{k+}^{\dagger}n_{\uparrow}d_{\downarrow}\right)$$

$$\times \hat{M}_{22}\left(i\alpha_{k'2}e^{-i\frac{\theta'}{2}}c_{k'-}^{\dagger}n_{\downarrow}d_{\uparrow} - ie^{i\frac{\theta'}{2}}\alpha_{k'1}c_{k'-}^{\dagger}n_{\uparrow}d_{\downarrow}\right)^{\dagger}$$

$$+ \sum_{kk'\theta\theta'} \tilde{V}_k^{\theta}\tilde{V}_{k'}^{*\theta'}\left(i\alpha_{k2}e^{-i\frac{\theta}{2}}c_{k-}^{\dagger}n_{\downarrow}d_{\uparrow} - ie^{i\frac{\theta}{2}}\alpha_{k1}c_{k-}^{\dagger}n_{\uparrow}d_{\downarrow}\right)$$

$$\times \hat{M}_{22}\left(\alpha_{k'1}e^{-i\frac{\theta'}{2}}c_{k'+}^{\dagger}n_{\downarrow}d_{\uparrow} + e^{i\frac{\theta'}{2}}\alpha_{k'2}c_{k'+}^{\dagger}n_{\uparrow}d_{\downarrow}\right)^{\dagger}$$

$$+ \sum_{kk'\theta\theta'} \tilde{V}_k^{\theta}\tilde{V}_{k'}^{*\theta'}\left(i\alpha_{k2}e^{-i\frac{\theta}{2}}c_{k-}^{\dagger}n_{\downarrow}d_{\uparrow} - ie^{i\frac{\theta}{2}}\alpha_{k1}c_{k-}^{\dagger}n_{\uparrow}d_{\downarrow}\right)$$

$$\times \hat{M}_{22}\left(i\alpha_{k'2}e^{-i\frac{\theta'}{2}}c_{k'-}^{\dagger}n_{\downarrow}d_{\uparrow} - ie^{i\frac{\theta'}{2}}\alpha_{k'1}c_{k'-}^{\dagger}n_{\uparrow}d_{\downarrow}\right)^{\dagger}.$$

(A.7)

In EQ (A.7) $\hat{M}_{22} = \frac{1}{E-H_{22}}$, we simplify using commutation algebra for operators of the form $\frac{1}{E-O_1}O_2$, where $[O_1,O_2]=cO_2$. This allows us to rewrite the expression as a power series and derive components in $H_{eff}$. In similar way for the component $\sum_{\zeta,\zeta'=\pm} H_{10}^{\zeta}\frac{1}{E-H_{00}}H_{01}^{\zeta'}$ replace $2 \rightarrow 0$ in above and resulting the matrix elements will be $\hat{M}_{00} = \frac{1}{E-H_{00}}$. From components above $\sum_{\zeta,\zeta'=\pm} H_{12}^{\zeta}\frac{1}{E-H_{22}}H_{21}^{\zeta'}$ and $\sum_{\zeta,\zeta'=\pm} H_{10}^{\zeta}\frac{1}{E-H_{00}}H_{01}^{\zeta'}$, we derive the following effective Hamiltonian.

$$H_{eff} = H_{11} + \sum_{kk'\zeta}^{\theta\theta'} M_{kk'\zeta}^{\theta\theta'}\left(e^{i\delta}\alpha_{k1}\alpha_{k'1}c_{k+}^{\dagger}d_{\uparrow}d_{\uparrow}^{\dagger}c_{k'+} + e^{-i\delta}\alpha_{k2}\alpha_{k'2}c_{k+}^{\dagger}d_{\downarrow}d_{\downarrow}^{\dagger}c_{k'+}\right.$$

$$\left. + e^{i\phi}\alpha_{k2}\alpha_{k'1}c_{k+}^{\dagger}d_{\downarrow}d_{\uparrow}^{\dagger}c_{k'+} + e^{-i\phi}\alpha_{k1}\alpha_{k2}c_{k+}^{\dagger}d_{\uparrow}d_{\downarrow}^{\dagger}c_{k'+}\right)$$

$$+ i\sum_{kk'\zeta}^{\theta\theta'} M_{kk'\zeta}^{\theta\theta'}\left(\alpha_{k1}\alpha_{k'1}e^{-i\phi}c_{k+}^{\dagger}d_{\uparrow}d_{\downarrow}^{\dagger}c_{k'-} - e^{-i\delta}\alpha_{k2}\alpha_{k'1}c_{k+}^{\dagger}d_{\uparrow}d_{\uparrow}^{\dagger}c_{k'-}\right.$$

$$\left. + e^{i\delta}\alpha_{k2}\alpha_{k'1}c_{k+}^{\dagger}d_{\downarrow}d_{\downarrow}^{\dagger}c_{k'-} - \alpha_{k2}\alpha_{k'2}e^{i\phi}c_{k+}^{\dagger}d_{\downarrow}d_{\uparrow}^{\dagger}c_{k'-}\right)$$

$$- i\sum_{kk'\zeta}^{\theta\theta'} M_{kk'\zeta}^{\theta\theta'}\left(\alpha_{k1}\alpha_{k'1}e^{i\phi}c_{k-}^{\dagger}d_{\downarrow}d_{\uparrow}^{\dagger}c_{k'+} - e^{i\delta}\alpha_{k2}\alpha_{k'1}c_{k-}^{\dagger}d_{\uparrow}d_{\uparrow}^{\dagger}c_{k'+}\right.$$

$$\left. + e^{-i\delta}\alpha_{k1}\alpha_{k'2}c_{k-}^{\dagger}d_{\downarrow}d_{\downarrow}^{\dagger}c_{k'+} - e^{-i\phi}\alpha_{k2}\alpha_{k'2}c_{k-}^{\dagger}d_{\uparrow}d_{\downarrow}^{\dagger}c_{k'+}\right)$$

$$+ \sum_{kk'\zeta}^{\theta\theta'} M_{kk'\zeta}^{\theta\theta'}\left(e^{i\delta}\alpha_{k1}\alpha_{k'1}c_{k-}^{\dagger}d_{\downarrow}d_{\downarrow}^{\dagger}c_{k'-} - e^{-i\phi}\alpha_{k2}\alpha_{k'1}c_{k-}^{\dagger}d_{\uparrow}d_{\downarrow}^{\dagger}c_{k'-}\right.$$

$$\left. - e^{i\phi}\alpha_{k1}\alpha_{k2}c_{k-}^{\dagger}d_{\downarrow}d_{\uparrow}^{\dagger}c_{k'-} + e^{-i\delta}\alpha_{k2}\alpha_{k'2}c_{k-}^{\dagger}d_{\uparrow}d_{\uparrow}^{\dagger}c_{k'-}\right),$$

(A.8)

where in the above, $\delta = \frac{\theta}{2} - \frac{\theta'}{2}$ and $\phi = \frac{\theta}{2} + \frac{\theta'}{2}$. It follows from EQ (A.8) that $H_{eff} = H_{eff}^{\dagger}$

since the summations over $k, k'$ are interchangeable. In EQ (A.8), the elements are

$$M_{kk'\zeta}^{\theta\theta'} = \hat{M}_{00} + \hat{M}_{22} = \frac{\tilde{V}_k^\theta \tilde{V}_{k'}^{\theta'}}{-\epsilon_d + \epsilon_{k\zeta}} + \frac{\tilde{V}_k^\theta \tilde{V}_{k'}^{\theta'}}{U + \epsilon_d - \epsilon_{k\zeta}}, \qquad \tilde{V}_k^\theta = \frac{V_k}{\sqrt{\beta\gamma k^3 \cos 3\theta}}.$$

In this model parameters $k$ and $\theta$ scaling differentiate conventional isotropic dispersion poorman scaling. Additionally, the parameter $\beta$ plays a role analogous to a magnetic-field coupled term, e.g. $BS_z$, but here it is associated with dispersion. *Note that the edge contribution, which appears solely in the cross terms—particularly $\vec{s} \times \vec{S}_{kk'}$. This explains why the model in EQ (16) exhibits topological properties.* We simplify by using the Abrikosov representation [31] for the impurity spin as $\mathbf{s} = \psi_d^\dagger \boldsymbol{\sigma} \psi_d$ and for the bath spins as $\mathbf{S}_k = \psi_k^\dagger \boldsymbol{\sigma} \psi_k$, where $\boldsymbol{\sigma}$ are the Pauli matrices,

$$\psi_d^\dagger = \begin{pmatrix} d_\uparrow^\dagger & d_\downarrow^\dagger \end{pmatrix}, \qquad \psi_k^\dagger = \begin{pmatrix} c_{k\uparrow}^\dagger & c_{k'\downarrow}^\dagger \end{pmatrix}.$$

Collecting all terms in EQ (A.8),

$$
\begin{aligned}
H_{eff}^1 = H_{11} + \sum_{kk'\zeta} M_{kk'\zeta}^{\theta\theta'} \Big( & (\alpha_{k1}\alpha_{k'1}e^{i\delta} + e^{-i\delta}\alpha_{k2}\alpha_{k'2})s_z S_{kk'}^z \\
& + (\alpha_{k1}\alpha_{k'1} + \alpha_{k2}\alpha_{k'2})\Big(e^{i\phi}s_- S_{kk'}^+ + e^{-i\phi}S_{kk'}^- s_+\Big) \\
& + \alpha_{k1}\alpha_{k'2}\underline{\Big(e^{i\phi}s_- S_{kk'}^z + e^{-i\phi}S_{kk'}^z s_+\Big)} \\
& + i(\alpha_{k1}\alpha_{k'1} - \alpha_{k2}\alpha_{k'2})\Big(e^{i\phi}s_- S_{kk'}^+ - e^{-i\phi}S_{kk'}^- s_+\Big) \\
& + \alpha_{k1}\alpha_{k'2}\underline{\underline{\Big(-ie^{i\delta}s_z S_{kk'}^- + ie^{-i\delta}S_{kk'}^+ s_z\Big)}}\Big).
\end{aligned}
\tag{A.9}
$$

Simplifying the doubly underlined terms leads to ADM-interaction. Substituting $(\alpha_{k1}, \alpha_{k2})$ as $\alpha_{k1} = \sqrt{\Delta + \beta k^3 \cos 3\theta}$, $\alpha_{k'2} = \sqrt{\Delta - \beta(k')^3 \cos 3\theta'}$ and in limit $k \to k'$ we get simplifications as $\alpha_{k1}^2 + \alpha_{k2}^2 = \Delta$, where the $\Delta = \sqrt{\beta^2 k^6 \cos^2 3\theta + \lambda^2 k^2}$, $\alpha_{k1}^2 - \alpha_{k2}^2 = \beta k^3 \cos 3\theta$ and $\alpha_{k1}\alpha_{k2} = k\lambda$. This model is then rewritten in the form of ADM interaction model as EQ (10). Analyzing the prefactors of $M_{kk'}^{\theta\theta'}$, which scale as $\frac{V_k}{\sqrt{\beta\gamma k^3 \cos 3\theta}} \frac{V_{k'}}{\sqrt{\beta\gamma(k')^3 \cos 3\theta'}}$, we can consider the scatterings by constructing a regular hexagon, as the $\cos 3\theta$ takes maximum values at the points of this hexagonal cell. For any given k point at band edges, these scatterings will be high-energy states and need to be integrated out. As we discussed in the main article, we proceed by constructing the new spinor as $\psi^\dagger = \begin{pmatrix} c_{k\frac{2\pi}{3}+}^\dagger & c_{k\frac{\pi}{3}-}^\dagger & c_{k-\frac{2\pi}{3}+}^\dagger & c_{k-\frac{\pi}{3}-}^\dagger \end{pmatrix}$; in this basis, the Hamiltonian may be written in a more general form discussed in the next section.

## B  Including the potential scattering

In the section 5 we discussed about adding potential scattering terms, Here we detail how one can use the lie matrices algebra to compute RG EQ's. We can expand the Eq (A.9) by expressing it in terms of magnitudes and operators in vector form, where the spin operator components are denoted with a hat symbol.

$$
\begin{aligned}
H_{eff}^{scat} = H_0 + H_{NH} + & \sum_{kk'} J_0 \Big( S_x \hat{\Sigma}_x + S_y \hat{\Sigma}_y + S_z \hat{\Sigma}_z \Big) + i\sum_{kk'} |\vec{J}_{k^3}| \Big( S_x \hat{\Sigma}_y - \hat{\Sigma}_x S_y \Big) \\
& + i\sum_{kk'} |\vec{J}_k| \Big( \big( S_y \hat{\Sigma}_z - \hat{\Sigma}_y S_z \big) + \big( S_x \hat{\Sigma}_z - \hat{\Sigma}_x S_z \big) \Big) \\
& + i\sum_{kk'} |\vec{g}_{2k}| \Big( \big( S_y \hat{\Gamma}_z - \hat{\Gamma}_y S_z \big) + \big( S_x \hat{\Gamma}_z - \hat{\Gamma}_x S_z \big) \Big),
\end{aligned}
\tag{B.1}
$$

$$H_{NH} = -\sum_{kk'} |\vec{g}_{1k^3}| \left(S_x \hat{\Omega}_y - \hat{\Omega}_x S_y\right) + \sum_{kk'} |\vec{g}_{2k^3}| \left(S_x \hat{\Gamma}_y - \hat{\Gamma}_x S_y\right)$$
$$- \sum_{kk'} |\vec{g}_{1k}| \left( \left(S_y \hat{\Omega}_z - \hat{\Omega}_y S_z\right) + \left(S_x \hat{\Omega}_z - \hat{\Omega}_x S_z\right) \right).$$

We use the diagrams [32–35] with all permutations of vertices using the following algebra for third order perturbation theory,

$$\begin{aligned}
[\Sigma_a, \Sigma_b] &= 4i\,\epsilon_{abc}\,\Sigma_c\,, \\
[\Gamma_a, \Gamma_b] &= 4i\,\epsilon_{abc}\,\Gamma_c\,, \\
[\Omega_a, \Omega_b] &= 4i\,\epsilon_{abc}\,\Omega_c\,, \\
[\Sigma_a\Sigma_b, \Sigma_c] &= 8i\,\epsilon_{abc}\,\mathcal{I} - 8\,\delta_{ab}\,\Sigma_c + 8\,\delta_{ac}\,\Sigma_b\,, \\
[\Gamma_a\Gamma_b, \Gamma_c] &= 8i\,\epsilon_{abc}\,\mathcal{I} - 8\,\delta_{ab}\,\Gamma_c + 8\,\delta_{ac}\,\Gamma_b\,, \\
[\Omega_a\Omega_b, \Omega_c] &= 8i\,\epsilon_{abc}\,\mathcal{I} - 8\,\delta_{ab}\,\Omega_c + 8\,\delta_{ac}\,\Omega_b\,, \\
[\Omega_a, \Sigma_b] &= 0\,, \\
[\Sigma_a\Omega_b, \Gamma_c] &= 0\,.
\end{aligned} \tag{B.2}$$

$$[\Omega_a\Sigma_b, \Sigma_c] = 4i\,\epsilon_{bcd} \begin{pmatrix} \sigma_a\sigma_d & \sigma_a\sigma_d \\ -\sigma_a\sigma_d & -\sigma_a\sigma_d \end{pmatrix},$$

Where in above EQ (B.2) $\Sigma_i = \begin{pmatrix} \sigma_i & \sigma_i \\ \sigma_i & \sigma_i \end{pmatrix}$, $\Omega_i = \begin{pmatrix} \sigma_i & 0 \\ 0 & -\sigma_i \end{pmatrix}$, $\Gamma_i = \begin{pmatrix} \sigma_i & 0 \\ 0 & \sigma_i \end{pmatrix}$ and $\Upsilon = \begin{pmatrix} \mathcal{I}_{2\times2} & 0 \\ 0 & \mathcal{I}_{2\times2} \end{pmatrix}$ these conventions used earlier [8]. The subscript $i$ refers a,b and c which are Pauli matrices. Using the algebra we derive RG EQ's as follows,

$$\begin{aligned}
\frac{dJ_0}{dl} &= J_0^2 + J_{k^3}^2 + J_k^2 + J_{k^3}J_k + J_k^2 J_0 + J_{k^3}^2 J_0 + J_0^3 - J_{k^3}g_{2k} - J_{k^3}g_{2k^3} - J_k g_{2k} \\
&\quad + g_{2k}^2 J_{k^3} + g_{2k^3}^2 J_k - J_k^2 g_{2k^3}\,,
\end{aligned}$$
$$\frac{dJ_{k^3}}{dl} = J_k^2 + J_0 J_{k^3} + J_k^2 J_{k^3} + J_0^2 J_{k^3} + J_{k^3}^3\,, \qquad \frac{dJ_k}{dl} = J_0 J_k + J_0^2 J_k + J_k J_{k^3}^2 + J_k^3\,, \tag{B.3}$$
$$\frac{dg_{1k^3}}{dl} = g_{1k^3}^3 - g_{1k}^2 g_{1k^3}\,, \qquad\qquad \frac{dg_{1k}}{dl} = g_{1k}^3 - g_{1k^2}^2 g_{1k}\,,$$
$$\frac{dg_{2k^3}}{dl} = -g_{2k^3}^3 + g_{1k^3}^2 g_{2k^3}\,, \qquad\qquad \frac{dg_{2k}}{dl} = -g_{2k}^3 + g_{1k}^2 g_{2k}\,.$$

From above, we identify various RG invariants as $\frac{g_{1k^3}}{g_{1k}} = m_1$, and we can notice that after adding the potential scattering terms, we still have the invariant $J_k^2 + J_k = m J_{k^3}$ and additionally $\frac{g_{2k}^2(g_{2k} + \sqrt{m_2})}{(g_{2k} - \sqrt{m_2})} = \frac{g_{2k^3}^2(g_{2k^3} + \sqrt{m_2/m_1})}{(g_{2k^3} - \sqrt{m_2/m_1})}$.

## B.1 Analytic solution of RG equations

A simplification of the EQ's (11) for $J_{k^3}$ and $J_k$ by eliminating $J_0$ yields the following:

$$\frac{dJ_{k^3}}{dl} - J_k^2 = \frac{J_{k^3}}{J_k}\frac{dJ_k}{dl}\,, \qquad \frac{1}{J_{k^3}}\frac{dJ_{k^3}}{dl} - \frac{J_k^2}{J_{k^3}} = \frac{1}{J_k}\frac{dJ_k}{dl}\,. \tag{B.4}$$

By performing a substitution $\frac{J_k^2}{J_{k^3}} = x$, which results in $\frac{dx}{dl} = 2\frac{J_k}{J_{k^3}}\frac{dJ_k}{dl} - \frac{J_k^2}{J_{k^3}^2}\frac{dJ_{k^3}}{dl}$, and simplifying the equation, we obtain the solution $J_k = \frac{1}{2} \pm \frac{1}{2}\sqrt{1 + 4mJ_{k^3}}$. We can derive the complete

solution by variable separation as follows:

$$J_0 \frac{dJ_0}{dl} - J_0^3 = n, \left( \frac{J_k^2 + J_k}{m} + J_k \frac{(J_k + 1)^2}{m^2} + J_k \right) \frac{dJ_k}{dl} = n. \tag{B.5}$$

The resulting solution will be in terms of two invariants, n and m. When we allow for complex solutions, we find that $J_k^* \to \frac{\sqrt{3} e^{i\frac{\pi}{3}}}{2}$, $J_{k^3}^* \to \frac{9 e^{i\frac{\pi}{3}}}{16}$, and $m = e^{i\frac{\pi}{3}}$, with $J_0^*$ having a $\tan^{-1}$ quantity that vanishes for $J_0^* = \pm 1$ in both ferromagnetic and antiferromagnetic cases as $n \to 1$. Please see EQ's in (B.6). Additionally, the logarithmic contribution to the scale vanishes when $J_0^* = -2$. These fixed points(complex) are obtained by incorporating the potential scattering terms.

$$\log \left( \frac{[m^2 + m(J_k - 1) + (J_k - 1)^2]^{3\gamma}}{J_k^{2\gamma}} \right) + \frac{(m-2)}{\sqrt{3}(m^2 - m + 1)} \tan^{-1} \left( \frac{m + 2J_k - 2}{\sqrt{3} m} \right) = n \log D_{eff}. \tag{B.6}$$

Solving first equation in (B.5) for $J_0$ as following,

$$\frac{\log \left( n^{2/3} + \sqrt[3]{n}(-J_0) + J_0^2 \right) - 2\log \left( \sqrt[3]{n} + J_0 \right)}{6\sqrt[3]{n}} \frac{-2\sqrt{3} \tan^{-1} \left( \frac{1 - \frac{2J_0}{\sqrt[3]{n}}}{\sqrt{3}} \right)}{6\sqrt[3]{n}} = \log D_{eff}. \tag{B.7}$$

In above equation $\gamma = \frac{-m}{6*(m^2 - m + 1)}$ and gamma diverges when $m = e^{\pm i\frac{\pi}{3}}$. As a result we expect complex fixed points and as exponent $\gamma$ diverge. A similarity exists between the RG EQ's for a single impurity in edge states and that of two impurities having DM interactions. To explore this connection, we address the two-impurity problem by simplifying the RG EQ's. We consider setting all couplings to zero, except for $J_0 \neq J_Y \neq K \neq 0$.

$$\frac{dJ_0}{dl} = J_0^2 + J_Y J_0 + K J_0, \qquad \frac{dJ_Y}{dl} = J_Y^2 + J_0^2 + K^2, \qquad \frac{dK}{dl} = K^2 + K J_Y. \tag{B.8}$$

By solving the EQ's for $J_0$ and $K$, we obtain the solution $J_0 = \frac{RK}{K-1}$, where $R$ is a constant or an invariant under renormalization. Using this solution, we proceed to solve for $J_Y$ and $K$ as follows:

$$J_Y \frac{dJ_Y}{dl} - J_Y^3 = R_1, \qquad \frac{dK}{dl} - K^2 = \frac{R_1}{\frac{R^2 K}{(1-K)^2} + K}. \tag{B.9}$$

Solution to the $J_Y$ equation can be written as follows:

$$\frac{\sqrt[3]{R_1}(-J_Y) + J_Y^2 + R_1^{2/3}}{(J_Y + \sqrt[3]{R_1})^2} e^{2\sqrt{3} \tan^{-1} \left( \frac{\frac{2J_Y}{\sqrt[3]{R_1}} - 1}{\sqrt{3}} \right)} = D_{eff}^{6R_1^{\frac{1}{3}}}. \tag{B.10}$$

For $K$-equation has higher order NL so we have solutions in two limits,

$$\left( \frac{\sqrt[3]{R_1}(-K) + K^2 + R_1^{2/3}}{\left( K + \sqrt[3]{R_1} \right)^2} \right) e^{+2\sqrt{3} \tan^{-1} \left( \frac{\frac{2K}{\sqrt[3]{R_1}} - 1}{\sqrt{3}} \right)} = D_{eff}^{6\sqrt[3]{R_1}}, \qquad \text{for } K \to \infty,$$

$$\frac{-\frac{R^2}{K-1} + R^2 \log(K-1) + \frac{1}{2}(K-1)^2 + K}{R_1} = \log D_{eff}, \qquad \text{for } K \to 0. \tag{B.11}$$

In a similar fashion, we will work on solving the $J_0$ equation with $J_Y$ in order to separate the solutions. For $R = R_1 = 1.0$, we will illustrate the RG flow in figure 13. We anticipate fixed points in different quadrants based on the sign chosen for these constants.

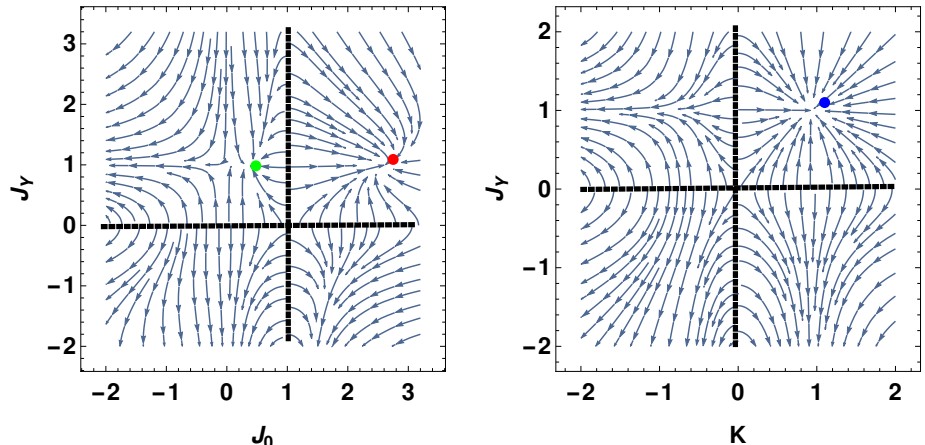

Figure 13: RG flow of two impurity show the FP can be in different quadrants with $R_1 = -1$ and $R = 1.0$. We get around the SP($(J_Y = 1.0, J_0 = 3.0)$) existing SR regime in couplings.

*RKKY Interaction:* For completeness, we briefly comment on the RKKY interaction in edge states, which arises from Fourier expansions of couplings. It's general form involves an integral over momentum and angular variables:

$$J_Y(r) \propto \int_k \frac{e^{ikR}}{k\beta} \oint \frac{q\, dt}{q^2 - (4t^3 - 3t)^2} + \sum_{kk'} \frac{|V_{ij}|}{k^2 - (k')^2} e^{ikr} e^{-ik'r}, \tag{B.12}$$

where $q = \frac{\sqrt{(k^2-\mu)^2-k^2}}{\beta k^3}$. The d contributions can be expressed as special functions, including elliptic integrals and transcendental terms. The NL contribution arises when $q \neq 0$. The second term yields standard Bessel- or elliptic-type RKKY oscillations [36, 37], while the first term reflects anisotropy/NL-induced deviations from flat-band physics.

## C  Impurity transport calculation

We follow the $\mathcal{T}_{kk'}$ formalism for the effective Kondo model in equation (10) to compute the relaxation time as follows,

$$\frac{1}{\tau} \propto \left(1 - 2J\tilde{g}_\zeta - 2J_{k^3}\tilde{g}_{\zeta k^3} - 2J_k\tilde{g}_{\zeta k}\right). \tag{C.1}$$

Where the $\zeta$ correspond to chiral index and takes vales $\zeta = \pm$ for each bands in effective model bands.

$$\tilde{g}_\zeta = \int_0^{2\pi} \int_0^\pi \int_0^{k_f} \frac{k^2 \sin\theta\, dk\, d\theta\, d\phi}{k^2 + \zeta\sqrt{k^6\beta^2 \cos^2 3\theta + \lambda^2 k^2} - \mu}, \tag{C.2}$$

where Solving the $\theta$ integral first, we get two pieces as follows,

$$\tilde{g}_\zeta = 2\pi \int_k k^2 \oint \frac{dt}{k^2 + \zeta\sqrt{k^6\beta^2(4t^3 - 3t)^2 + \lambda^2 k^2} - \mu}, \tag{C.3}$$

we rationalize the above integral and write as the following by introducing $q = \frac{\sqrt{(k^2-\mu)^2 - \lambda^2 k^2}}{\beta k^3}$,

$$\tilde{g}_\zeta = 2\pi \int_k \frac{1}{k} \oint \frac{q + \bar{\zeta}\sqrt{(4t^3 - 3t)^2 + \frac{\lambda^2}{\beta^2 k^4}}\, dt}{q^2 - (4t^3 - 3t)^2} = \int_k \frac{\pi}{kq}\left(\sum_{res} f(t,q) + \sum_{res} f(t,-q)\right). \tag{C.4}$$

For finding these residues, we will use the roots of the cubic EQ's $\beta(4t^3 - 3t) \pm q = 0$. We collect positive q roots and negative as follows for doing contour integrals,

$$t^\zeta = \begin{cases} \frac{1}{2}\left(\sqrt[3]{\sqrt{q^2-1}+\zeta q} + \frac{1}{\sqrt[3]{\sqrt{q^2-1}+\zeta q}}\right), \\ -\frac{1}{4}\left(1-i\sqrt{3}\right)\sqrt[3]{\sqrt{q^2-1}+\zeta q} - \frac{1+i\sqrt{3}}{4\sqrt[3]{\sqrt{q^2-1}+\zeta q}}, \\ -\frac{1}{4}\left(1+i\sqrt{3}\right)\sqrt[3]{\sqrt{q^2-1}+\zeta q} - \frac{1-i\sqrt{3}}{4\sqrt[3]{\sqrt{q^2-1}+\zeta q}}. \end{cases} \tag{C.5}$$

Where in above we have $\zeta = \pm$, Similarly, the other contributions are as follows,

$$\tilde{g}_{\zeta k^3} = 2\pi\beta \int_k k^2 \oint \frac{((4t^3-3t)^2)(q+\bar{\zeta}r)}{q^2-(4t^3-3t)^2}dt, \qquad \tilde{g}_{\zeta k} = \int_k \oint \frac{(q+\bar{\zeta}r)}{q^2-(4t^3-3t)^2}dt, \tag{C.6}$$

where in (C.6) $r = \sqrt{(4t^3-3t)^2 + \frac{\lambda^2}{\beta^2 k^4}}$, after summing over the $\zeta = \pm$ bands we get following,

$$\tilde{g} = 2\pi \int_k \frac{1}{k} \oint \frac{q\,dt}{q^2-(4t^3-3t)^2},$$
$$\tilde{g}_{k^3} = 2\pi\beta \int_k k^2 \oint \frac{(4t^3-3t)^2 q\,dt}{q^2-(4t^3-3t)^2}, \tag{C.7}$$
$$\tilde{g}_k = \int_k \oint \frac{q\,dt}{q^2-(4t^3-3t)^2}.$$

After contour integrals each contributes as $\oint \frac{q\,dt}{q^2-(4t^3-3t)^2} = q$ hence sum of all $g_\zeta$ contribution as following,

$$\tilde{g}(\mu) = \int \left(\beta q k^2 - \beta q^4 k + \frac{q}{k} - q\right)dk. \tag{C.8}$$

We set $\epsilon \approx k^2$ and the density of states in 3D as $\rho(\epsilon) \approx \sqrt{\epsilon}$ then above integral yields,

$$\tilde{g}(\mu) = \int \left(\frac{\sqrt{(\epsilon-\mu)^2-\lambda^2\epsilon}}{\sqrt{\epsilon}} - \frac{((\epsilon-\mu)^2-\lambda^2\epsilon)^2}{\beta^3\epsilon^{6-\frac{1}{2}}} + \frac{\sqrt{(\epsilon-\mu)^2-\lambda^2\epsilon}}{\beta\epsilon^2} - \frac{\sqrt{(\epsilon-\mu)^2-\lambda^2\epsilon}}{\beta\epsilon^{\frac{3}{2}}}\right)d\epsilon. \tag{C.9}$$

Where in above $\mu$ is the chemical potential can take any values around Fermi energy. We perform this above integral exactly in terms of special functions to extract contribution to $\frac{1}{\tau}$, which indeed scales with RG invariant in the elliptic functions.

$$\tilde{g}_\epsilon \propto \frac{P_1(\epsilon) + P_{\frac{3}{2}}(\epsilon) + T_1(\epsilon) + E_1(\epsilon) + E_2(\epsilon) + \log(\epsilon)}{\beta^3},$$

$$P_1(\epsilon) = -\frac{3\mu^4 - 16\mu^3\epsilon + 4\mu^2\epsilon(9\epsilon-2)}{12\epsilon^4}$$
$$-\frac{2\epsilon^2\left(2\epsilon\left(\beta^2\sqrt{(\mu-\epsilon)^2-\epsilon}\left(2\sqrt{\epsilon}(\beta-\beta\epsilon+3)+3\right)-6\right)+3\right)}{12\epsilon^4},$$

$$P_{\frac{3}{2}}(\epsilon) = \frac{8\mu\epsilon^2\left(2\beta^3\epsilon^{3/2}\sqrt{(\mu-\epsilon)^2-\epsilon}-6\epsilon+3\right)}{12\epsilon^4},$$

$$T_1(\epsilon) = \beta^2\tanh^{-1}\left(\frac{-2\mu+2\epsilon-1}{2\sqrt{(\mu-\epsilon)^2-\epsilon}}\right) + \frac{(2\mu+1)\beta^2\coth^{-1}\left(\frac{2\mu\sqrt{(\mu-\epsilon)^2-\epsilon}}{2\mu(\mu-\epsilon)-\epsilon}\right)}{2\mu},$$

$$E_1(\epsilon) = \frac{i\beta^2\epsilon\sqrt{\frac{-2\mu+r+2\epsilon-1}{\epsilon}}\sqrt{-\frac{2\mu+r-2\epsilon+1}{\epsilon}}\left(AE\left(i\sinh^{-1}\left(\frac{\sqrt{-2\mu-r-1}}{\sqrt{2}\sqrt{\epsilon}}\right)\Big|\frac{2\mu-r+1}{2\mu+r+1}\right)\right)}{3\sqrt{2}\sqrt{-2\mu-r-1}\sqrt{(\mu-\epsilon)^2-\epsilon}},$$

$$A = (2\mu+r+1)(2\mu\beta+\beta+6),$$

$$E_2(\epsilon) = -\frac{B\left((r+2\mu(r+2)+1)\beta+6r\right)F\left(i\sinh^{-1}\left(\frac{\sqrt{-2\mu-r-1}}{\sqrt{2}\sqrt{\epsilon}}\right)\Big|\frac{2\mu-r+1}{2\mu+r+1}\right)}{3\sqrt{2}\sqrt{-2\mu-r-1}\sqrt{(\mu-\epsilon)^2-\epsilon}},$$

$$B = i\beta^2\epsilon\sqrt{\frac{-2\mu+r+2\epsilon-1}{\epsilon}}\sqrt{-\frac{2\mu+r-2\epsilon+1}{\epsilon}}, \tag{C.10}$$

where $r = \sqrt{4\mu+1}$ in above equation (C.10). In the above solution, F is an incomplete elliptic function of the first kind, and E is an elliptic function of the second kind. P and T are polynomial and transcendental functions, respectively.

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
