# Peer review of "Anderson Impurities In Edge States with Nonlinear and Dissipative Perturbations"

_SciPost Physics, doi:SciPost Phys. 19, 036 (2025)_

## Round 3 · Author Response

Response to Referees:

We thank both Referee 1 (Pradip Kattel) and Referee 2 for their thoughtful and detailed feedback. We have carefully revised the manuscript in response to their comments. Below, we provide a point-by-point reply to each referee, outlining the changes made and the clarifications added.

Response to Referee 1: Pradip Kattel Strengths acknowledged: We are grateful for your recognition of the novelty of our approach, including the treatment of higher-order momentum terms, the emergence of exceptional points (EPs), and the combined use of RG, transport, and Fock-space diagonalization techniques.

1) “The paper is extremely heavy on formal derivations... difficult to follow.”

Response: We have reorganized the manuscript to improve readability:Sections are now streamlined to clearly separate derivations from physical interpretation. New figures (e.g., Figs. 2 - 7) visualize RG flow, condition numbers, and EP signatures, making the key ideas more accessible. NH (non-Hermitian) terms are explicitly labeled in equations (e.g., HpotNH) to avoid confusion.

2) “Explain how the model with complex-valued coupling arises from Eq. 1 or Eq. 2.”

Response:In Section 3 and Appendix 10.1, we now clearly show that the complex-valued effective couplings emerge from a Hermitian Anderson impurity model via projection and 2D (k,\theta) scaling which is different from isotropic flat band poorman scaling. This projection includes a nonlinear hybridization function derived from a bath with momentum-dependent dispersion, leading to anisotropic and complex-valued RG couplings in the effective spin model.

3) “Explain why should the low-energy limit of a Hermitian model exhibit non-Hermiticity?”

Response:We emphasize throughout the revised manuscript (esp. Sections 1–3) that the original model is Hermitian, and that non-Hermiticity emerges effectively in the low-energy impurity subspace due to:Bath integration that produces complex self-energy contributions,Anisotropic and nonlinear hybridization functions that break reciprocity, and Complex-valued couplings generated under RG flow.This is analogous to lifetime effects or complex self-energies arising in interacting systems. We have also clarified that we do not assume a non-Hermitian model from the outset.

4) Citations missing and formatting issues

Response:All broken citations have been fixed. The references to Rashba-type models, Bethe ansatz studies, and non-Hermitian CFT are now correctly cited. We have also corrected typographic issues, spacing errors, and inconsistent punctuation.

Response to Referee 2: Anonymous

We thank the referee for highlighting the originality and relevance of our results concerning EPs and RG dynamics in nonlinear impurity systems.

1) “Justification of the original Hamiltonian and emergence of non-Hermiticity should be clarified.”

Response:In Sections 2–3 and Appendix 10.1, we provide a systematic derivation of the effective spin Hamiltonian from a fully Hermitian Anderson impurity model. The effective non-Hermitian terms are shown to arise from:Complex hybridization functions induced by nonlinear bath dispersion,Projection into the impurity sector, and RG flow that dynamically generates complex-valued couplings. We clarify that we do not assume a non-Hermitian Hamiltonian a priori, and that the emergent NH structure is a consequence of the RG and bath geometry.

2) “Origin and definition of exceptional points (EPs) is unclear.”

Response:We now provide an explicit definition of EPs in the Introduction, emphasizing their identification via coalescing eigenvalues and eigenvectors (non-diagonalizability). In Section 4, we derive conditions (e.g., Eqs. 29 and 31) under which EPs arise in both the RG flow and the effective Hamiltonian. Figures 5–7 visually confirm the EP behavior via spectral collapse and condition number divergence.

3) “Clarify which parts are Hermitian and which are not.”

Response:We now explicitly distinguish Hermitian and non-Hermitian components: NH contributions are labeled (e.g., HpotNH),Symmetry analysis (Section 6) uses metric operators η to show pseudo-Hermiticity and PC symmetry,Hermitian terms are separated from dissipative terms in both equations and figures. Additionally, we now refer to the appendices consistently and revise presentation throughout to make the manuscript more readable.

General Revisions Summary All RG equations, effective Hamiltonians, and symmetry transformations have been reorganized for clarity. New figures added to highlight RG flow structure, spiral points, and EP emergence. Condition number analysis and matrix block symmetries now support the topological interpretation of EPs. Transport analysis (Section 7) shows scaling collapse consistent with EP-related transitions.

Closing Remarks, We thank both referees for their insightful critiques, which helped us significantly improve the presentation, clarify our main physical claims, and rigorously justify the emergence of non-Hermiticity in our setting. We hope the revised version addresses all concerns. Sincerely, Vinayak M. Kulkarni (on behalf of all co-authors)

---

## Round 3 · List of Changes

List of Changes

1) Clarified emergence of non-Hermiticity from a fully Hermitian model (Sections 2–3 and Appendix 10.1). The projection procedure and complex-valued hybridization function are now explicitly shown to generate effective non-Hermitian terms dynamically.

2) Defined exceptional points (EPs) explicitly in the Introduction and derived the conditions for EPs analytically in Section 4 (Eqs. 16 - 19). Figures 2–5 now visualize EP emergence via condition number and spectral flow.

3) Improved notation and clarity: Non-Hermitian terms are labeled with superscripts/subscripts (e.g., HpotNH) and the distinction between Hermitian and non-Hermitian sectors is made clear throughout the text.

4)Reorganized presentation: Separated lengthy derivations from physical discussion. Each section now begins with a brief summary of goals and ends with interpretation of results.

5) Expanded explanation of RG flow in the complex plane, including the origin of spiral fixed points (SPs) and sign-reversion (SR) regimes. Citations fixed: All placeholder references (“[?]”) are now properly included and formatted. New references were added where necessary.

6) Appendix references corrected: All appendices are now consistently labeled and cited (e.g., Appendix 10.1, 10.2), matching the main text.

7) New figure added:Figure summarizing coupling structure in the two-impurity model (Fig. 8).

8) Typographic and formatting improvements: Corrected punctuation, spacing, and alignment issues throughout the manuscript.

---

## Editorial Decision

published